



# Quantification of allochthonous and autochthonous organic carbon in large and shallow Lake Wuliangsu based on distribution patterns and δ¹³C signatures of *n*-alkanes

Qingfeng Zhao[1], Aifeng Zhou[2], Yuxin He[1*]

[1] Organic Geochemistry Unit, Key Laboratory of Geoscience Big Data and Deep Resource of Zhejiang Province, School of Earth Sciences, Zhejiang University, Hangzhou, 310058, China

[2] Key Laboratory of Western China's Environmental Systems (Ministry of Education), College of Earth and Environmental Sciences, Lanzhou University, Lanzhou, 730000, China

*Correspondence to*: Yuxin He (yxhe@zju.edu.cn)

**Abstract:**

Identification and quantification of allochthonous and autochthonous organic carbon (OC) are crucial for the interpretation of burial behaviors of sedimentary OC of shallow lakes under anthropogenic interferences. In this study, we analyzed distribution patterns and δ¹³C signatures of mid- and long-chain *n*-alkanes on various types of surface samples from a typically large and

shallow Lake Wuliangsu in the Hetao Irrigation District. The results indicate that *n*-alkanes among submerged macrophytes, emergent plants, and riverine soil show unique distribution patterns and δ¹³C signatures, supporting the practicability of quantification on OC of these sources by end-member mixing models. Introducing the δ¹³C values of *n*-alkanes into the end-member mixing models could effectively reduce the potential error derived from end-member determination on *n*-alkane distribution patterns and OC degradation. The model results suggest that the riverine sourced OC from the main channel to

Lake Wuliangsu has settled down during the southward migration process. Therefore, Lake Wuliangsu serves as an important trap and sink for the allochthonous OC from the Upper Yellow River Reaches. The model results also show a predominate contribution from the autochthonous OC to Lake Wuliangsu (mostly >85%), with open-water areas dominated by submerged macrophytes and the rest of areas by emergent plants, largely modulated by water transparency, water depth, and nutrient concentrations. Together with previously published tetraether results, we further proposed that areas dominated by submerged

macrophytes might be more favorable for heterotrophic anaerobic bacteria and methanogenic archaea, largely due to active recycling processes for the labile OC derived from submerged macrophytes.

**Keywords:** Allochthonous *vs*. autochthonous OC, Lake Wuliangsu, *n*-alkanes, distribution patterns, δ¹³C values, end-member mixing model

## 1. Introduction

Shallow lakes (max water depth < 4 m), accounting for the largest area of lakes globally (Downing et al. 2006; Verpoorter et al., 2014), are very sensitive to climatic variations and land use and land cover (LULC) change (Sivakumar et al., 2005;

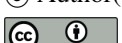



Harpenslager et al., 2022). As the largest lake ecosystem in the reaches of the Yellow River, Lake Wuliangsu is a typical semi-arid shallow lake that has undergone strong eutrophication and paludification due to rapid industrialization and urbanization

over the last decades. For instance, substantial nutrients input from agricultural and industrial activities have triggered eutrophication of this lake, resulting in stronger organic carbon (OC) burial and severer damage to the lake ecosystem (He et al., 2015, 2022; Shen et al., 2016; Sun et al., 2019). Meanwhile, autochthonous bioproductivity accumulation, allochthonous detritus siltation, and water loss together have resulted in intensive paludification of Lake Wuliangsu (Duan et al., 2005; Lü et al., 2008; He et al., 2015). Therefore, knowledge of OC burial behavior of Lake Wuliangsu under anthropogenic interferences

would provide an important reference for sustainable management practices of shallow lakes regionally and globally.

Generally, allochthonous OC (terrestrial input) and autochthonous OC (primary bioproductivity) in shallow lake systems show different characteristics on perspectives of source and sink behaviors (Alahuhta et al., 2014). On one hand, Lake Wuliangsu receives allochthonous materials from the surrounding catchment through local precipitation and discharge, and the Upper Yellow River Reaches via the western inlet channels (Wu et al., 2013). On the other hand, the *in-situ* bioproductivity

of Lake Wuliangsu is mainly contributed by submerged macrophytes and emergent plants (Sun et al., 2006; Tian et al., 2011; Fu et al., 2013; Zhang. 2017; Du et al., 2022; Ni et al., 2022), which show different ecological functions and OC burial characteristics (Botrel and Maranger, 2023). The stems and leaves of submerged macrophytes can adsorb heavy metals from water, and fix pollutants through their roots to improve the quality of lake water bodies (Zhu et al., 2018). Also, the outspread of submerged macrophytes would increase the carbon dioxide ($CO_2$) and methane ($CH_4$) release through the methanogenic

decomposition of plant exudations and debris (Li et al., 2019; He et al., 2023). Alternatively, emergent plants have a greater capacity for nutrient accumulation, and could significantly impact the biodiversity of macroinvertebrates within the vegetated area (Zhao et al., 2012). Emergent plants also accelerate the paludification of shallow lakes, largely due to their stronger evapotranspiration and faster siltation (Zhou and Zhou, 2009). Therefore, identification and quantification of OC from these sources would be very crucial to the interpretation of burial and decomposing behaviors of sedimentary OC in shallow lakes.

Over the past decades, many studies have attempted to evaluate the relative contributions of allochthonous and autochthonous OC to Lake Wuliangsu. For instance, previous studies have suggested a major contribution of autochthonous OC to Lake Wuliangsu, as evidenced by similar spatial distribution between the autochthonous biomass and the sedimentary TOC values (Sun et al., 2006; Tian et al., 2011; Fu et al., 2013). Also, *in-situ* biomasses of both submerged macrophytes and emergent plants have been directly evaluated by remote sensing techniques and on-site investigations (Zhang, 2017; Du et al.,

2022; Ni et al., 2022). The results have shown that half of the lake area is covered by emergent plants (known as vegetation-covered areas), while the other half is covered by submerged macrophytes and clear water (known as open-water areas, Zhang, 2017). Alternatively, the amount of phytoplankton is relatively lower compared to the widespread emergent plants and submerged macrophytes (Ni et al., 2022). In terms of seasonality, the biomass of submerged macrophytes is higher in September than in July (Du et al., 2022), while the biomass of emergent plants is higher in summer (June to August) than in



winter (January), largely due to artificial harvesting (Yang et al., 2009). Unfortunately, these studies have hardly quantified the

individual OC burial behaviors of both submerged macrophytes and emergent plants, let alone their comparison with the

allochthonous OC, asking for new methods to further evaluate the relative contributions of different OC sources to Lake

Wuliangsu.

    *n*-Alkanes have been proven to be useful in differentiating OC sources in lake system (Ficken et al., 2000; Castañeda and

Schouten, 2011). Generally, *n*-alkanes of both terrestrial vascular plants and emergent plants are characterized by the

predominance of long-chain homologues ($n$-$C_{27}$, $n$-$C_{29}$, and $n$-$C_{31}$). Alternatively, submerged plants mainly display a high

abundance of mid-chain homologues ($n$-$C_{23}$ and $n$-$C_{25}$), distinctive from both terrestrial and emergent plants (Ficken et al.,

2000; Aichner et al., 2010; Liu and Liu, 2016; Andrae et al., 2020). *n*-Alkanes have been successfully applied into Lake

Wuliangsu on a sedimentary core for the reconstruction of ecological dynamics since the Industrial Era (He et al., 2015).

However, without proper identification of *n*-alkane from modern samples as end-members, applications of *n*-alkanes might be

slightly hindered by overlaps in *n*-alkane compounds from various OC sources (Liu and Liu, 2016; Dion-kirschner et al., 2020).

Firstly, the contributions of long-chain *n*-alkanes from submerged macrophytes (Liu et al. 2018), as well as mid-chain *n*-

alkanes from emergent plants and terrestrial plants (Sinninghe Damsté et al., 2011) should be taken into consideration.

Secondly, emergent plants and terrestrial vascular plants both yield similar long-chain *n*-alkane predominance, usually

resulting in inevitable difficulty in separating both sources in shallow lakes (Liu and Liu, 2016). Thirdly, subsequent

degradation of *n*-alkanes during and after deposition might potentially increase the average carbon chain length, bringing

uncertainties into the application of *n*-alkane distribution patterns in differentiating OC sources (Van Beilen et al., 2003; Wang

et al., 2019).

    Compound specific $\delta^{13}$C values of *n*-alkanes serve as another independent tool for identification and quantification of

allochthonous and autochthonous OC (Sinninghe Damsté et al., 2011; Holtvoeth et al., 2019; Struck et al., 2019), considering

large $\delta^{13}$C differences in *n*-alkanes among different sources (Freeman et al., 1994) and little isotopic fractionation during photo-

oxidation and microbial degradation of *n*-alkanes (Ahad et al., 2011; Hyun et al., 2017). For autochthonous OC, submerged

macrophytes show relatively positive $\delta^{13}$C values of *n*-alkanes (e.g., –26.3‰ to –13.3‰, Aichner et al., 2010), while emergent

plants show much more negative $\delta^{13}$C values in *n*-alkanes (e.g., –35.3‰ to –28.7‰, Duan et al., 2011; Ho et al., 2015). The

$\delta^{13}$C values of soil OC are dependent on the natures of photosynthesis pathways of the sourced terrestrial plants ($C_3$ *vs.* $C_4$, –

39.0‰ to –32.0‰ *vs.* –25.0‰ to –18.0‰, Collister et al., 1994). A previous study on a sedimentary core in Lake Wuliangsu

suggested different photosynthesis pathways between mid- and long-chain *n*-alkanes, considering that $\delta^{13}$C values of mid-

chain *n*-alkanes were up to 7–9‰ more positive than those of long-chain ones in the same sample (He et al., 2015). They also

found that $\delta^{13}$C values are more depleted in both mid- and long-chain *n*-alkanes in Lake Wuliangsu after the 1990s than those

during the 1950s–1970s, tentatively signaling a shift in the OC sources upon time sequence. Accordingly, with the assistance

of compound-specific $\delta^{13}$C analysis, sources of *n*-alkanes in Lake Wuliangsu can be further elucidated.

In this study, we analyzed the distribution patterns and $\delta^{13}C$ values of mid- and long-chain $n$-alkanes from modern aquatic plants and surface sediments collected from the large and shallow Lake Wuliangsu. We aim to identify the distribution patterns and $\delta^{13}C$ characteristics of $n$-alkanes from major contributors to the sedimentary OC of Lake Wuliangsu, and develop

quantitative/semi-quantitative methods to calculate the relative contributions to sediments from different OC sources based on the end-member mixing models. We also discussed whether introducing the $\delta^{13}C$ values of $n$-alkanes into the end-member mixing models could effectively reduce the potential uncertainty. With the new end-member mixing models, we probed the spatial pattern and the controlling factors of allochthonous and autochthonous OC of Lake Wuliangsu under anthropogenic interferences, which would provide an important reference for sustainable management practices of shallow lakes regionally

and globally.

## *2. Materials and methods*

### 2.1 Geographical background

Lake Wuliangsu (40°47′–41°03′N, 108°43′–108°57′E) located in the Upper Yellow River Reaches, China (Figure. 1),

covers a surface area of 371 km². It is an alkaline (pH of 9.04) and fresh lake (salinity of 2.48 g/L, Ma et al., 2013), with a mean depth of 1–1.5 m and a maximum depth of 2.5–3.0 m (Song et al., 2019). According to the meteorological data from Urad Qianqi station, mean annual air temperature, precipitation, and potential evapotranspiration are 7.5 ℃, 225, and 2140 mm, respectively (data from http://www.ntsg.umt.edu), resulting in strong water loss (Sun et al., 2013). Lake Wuliangsu is surrounded by Lang Mountains to the north, Wula Mountain to the east, and the Yellow River floodplain to the west and the

south (Wu et al., 2013). It mainly receives discharge from the Hetao Irrigation District via six irrigation channels from the west bank, with the main channel (also known as the Wujia River) contributing ~90% of the total materials (Yang et al., 2019). Annual agricultural nutrient input into Lake Wuliangsu from the surrounding farm areas is ~65.75 t/yr (Ni et al., 2022). Dense emergent plants and submerged macrophytes are widely distributed across the Lake Wuliangsu under hypereutrophic conditions. Almost half of areas is covered with emergent plants (e.g., *Phragmites communis*), whereas submerged

macrophytes (e.g., *Potamogeton pectinatus*) become the predominant aquatic vegetation in the open-water area (Lü et al., 2008). Aquatic litters from submerged macrophytes and emergent plants quickly accumulated at the lake bottom at the rate of ~9–13 mm/yr, leading to server paludification for Lake Wuliangsu (Yu et al., 2007).

### 2.2 Sample collection

In July 2019, lake surface sediment samples (< 3 cm) were collected from 14 different locations within Lake Wuliangsu (sites S1–S14, Figure. 1).  For the 14 surface sediments, samples S3, S11, S12, S13, and S14 were collected in areas mainly covered by submerged macrophytes (open-water area), while samples S1, S2, S4, S5, S6, S7, S8, S9, and S10 were in areas mainly covered by emergent plants (vegetated area). Seven lake shore surface sediments (<3 cm) around the lake (sites SS1–



SS7, Figure. 1) were also collected. One riverine surface sediment (<5 cm) was collected at the end section of the main

irrigation channel (site WJ, Figure.1) to track riverine OC input. Furthermore, aquatic plants of *Phragmites communis* and

*Potamogeton pectinatus* were collected for the OC source appointment of emergent plants and submerged macrophytes. All

samples were preserved in a –20 ºC refrigerator immediately after being collected in the field and were freeze-dried once taken

back to the laboratory.

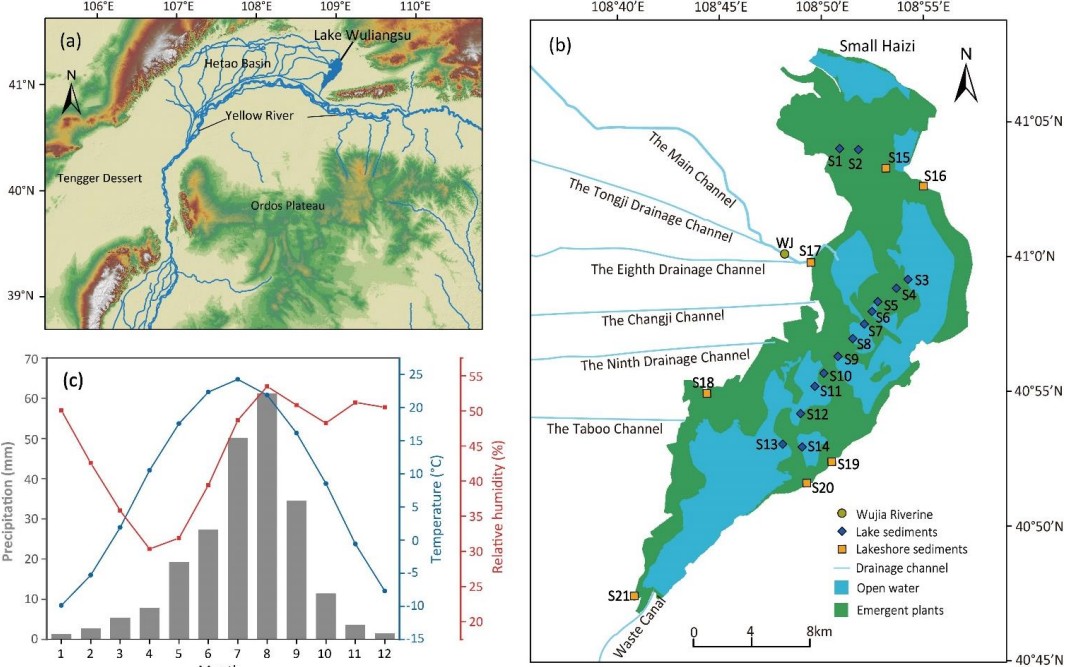

**Figure 1. Location and setting. (a) Map showing the location of study site Lake Wuliangsu (blue area) in the northern Chinese Loess Plateau. (b) the geophysical showing the hydrological condition and sample sites, the dark blue rhombus and orange square indicate the sampling sites within the lake and lakeshore, respectively. And the green and blue regions are *Phragmites australis* growth zones and open-water zones, (c) mean monthly precipitation, temperature, and relative humidity (1981-2015) from Urad Qianqi station, China (data from China Meteorological Administration https://data.cma.cn/), which is closed to Lake Wuliangsu.**


### 2.3 Bulk organic carbon analyses

Samples for total organic carbon (TOC) analysis were treated with 6N HCl at 60 ºC, heated for 2 h to remove carbonates, and subsequently rinsed with deionized water to neutral. The carbonates-free samples were then dried at 40 ºC and further analyzed for TOC contents on a Euro Vector EA 3000 Elemental Analyzer. The standard samples were analyzed between every

six samples, and the 1 σ precision of replicate analysis is ±0.02% for TOC analysis.

### 2.4 *n*-Alkane analysis

All the freeze-dried sample (~10 g sediments and ~1 g plant samples) were extracted ultrasonically with the solves of

dichloromethane/methanol (9: 1, v/v). After drying with a gentle $N_2$ stream, the extracted lipids were then saponified with 6%

KOH in methanol and further extracted with *n*-hexane to get the neutral fraction. The apolar fraction containing *n*-alkanes was

further obtained from the neutral fraction by silica gel liquid chromatography using *n*-hexane as eluent.  Quantification was

obtained using an Agilent 7890A gas chromatograph (GC) equipped with a flame ionization detector (FID). The GC was

equipped with a DB-1MS capillary column (60 m×0.32 mm ×0.25 μm film thickness). Each sample was injected in splitless

mode, with a GC inlet temperature of 295 ºC and a flow rate of 1.0 mL/min. The oven temperature was initially 60 ºC (held

for 2 min), then increased to 300 ºC at 4 ºC /min, and finally held for 30 min. Individual *n*-alkane peak areas were compared

with external standards (*n*-tetracosane-d$_{50}$) with known concentrations to calculate the concentrations of *n*-alkanes. Analytical

uncertainty for concentrations of *n*-alkanes would be less than 5%, and the distribution pattern of the *n*-alkanes are only slightly

impacted by the analytical uncertainty. *n*-Alkane-based $P_{aq}$ (proportion of aquatic plants) and CPI (carbon preference index)

proxies were calculated using the following equations provided in the Supplementary Material.


**2.5 Compound-specific δ¹³C analysis**

The neutral fraction containing *n*-alkanes was further separated into *n*-alkanes and branched/cyclic alkanes by urea

adduction. Compound-specific δ¹³C values of *n*-alkanes of lake sediments were measured in Thermo-Fisher Scientific Trace

GC coupled with a MAT 253 isotope ratio mass spectrometry (IRMS). The GC was also equipped with a DB-1MS capillary

column (60 m×0.32 mm×0.25 μm film thickness). Helium with a flow rate of 1.4 mL/min was used as carrier gas. The GC

oven temperature was initiated at 80 °C (held for 2 min), then increased to 300 ºC by 3 ºC/min, and finally held for 30 min. An

external standard consisting of a mixture of *n*-$C_{16}$ to *n*-$C_{30}$ alkanes with known isotopic values (B4, Indiana University) was

measured routinely to ensure the data quality. The carbon isotope ratios are expressed in ‰ relative to the V-PDB standard,

and the 1 σ precision of the replicate measurement on standard samples was < ±0.3‰.


***3. Results***

**3.1 TOC values**

The averaged TOC values for lake sediments, lakeshore sediments, and riverine sediment are 3.30% (0.71%–7.19%,

n=14), 0.78% (0.28%–1.30%, n=7), and 0.20%, respectively (Figure. 2a). Relatively higher TOC values in the lake sediments

than those in the lakeshore and riverine sediments suggest an additional supply of autochthonous materials of higher TOC

values (33.73% and 37.08% for submerged macrophytes and emergent plants). For the lake sediments, the TOC value was the

highest in the central section of the lake (site S12, 7.19%) and the lowest near the main channel (site S4, 0.71%). Moreover,

the TOC values of samples at the open-water area are slightly higher than those in the shallower vegetated area (avg. 5.53%

*vs.* 3.02%). Therefore, the sedimentary TOC values might be impacted by both the dilution of the terrestrial detritus and the

*in-situ* ecological conditions.



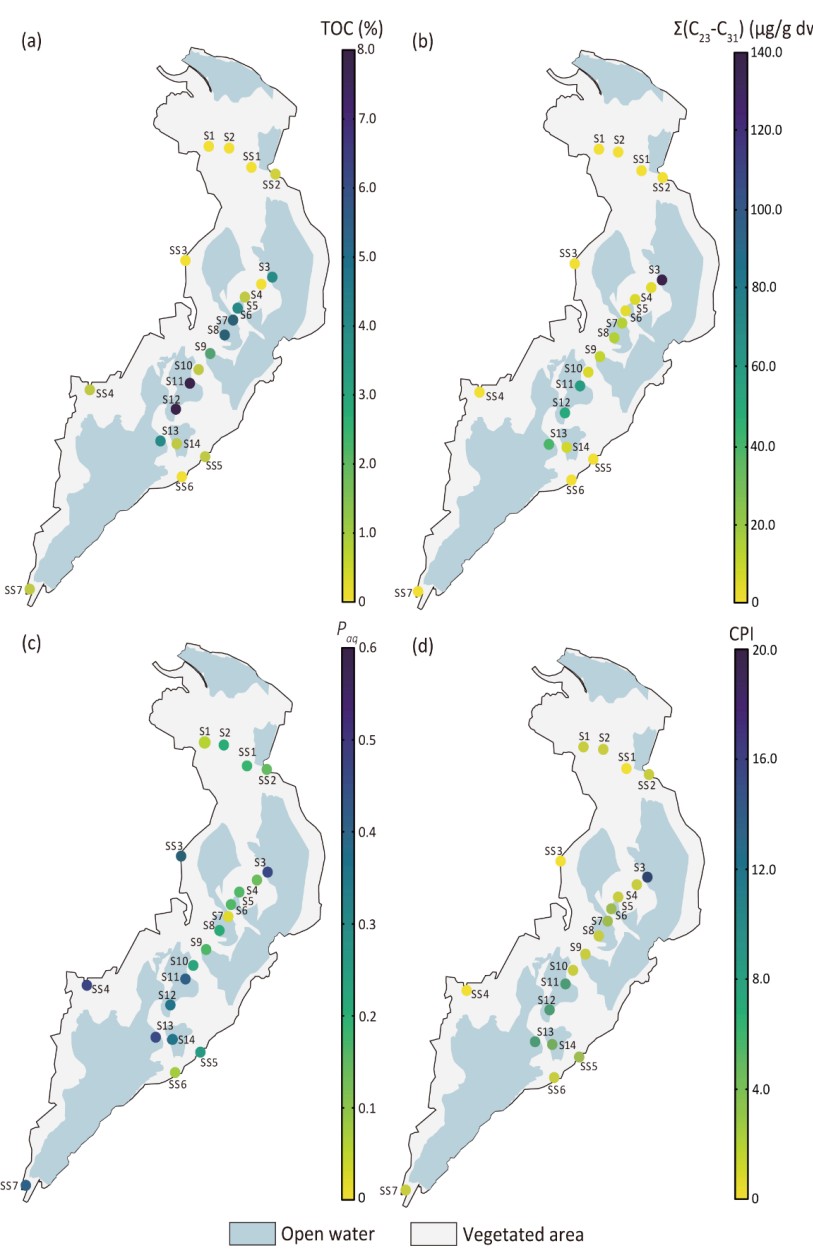

**Figure 2. Spatial distribution of (a) TOC values, (b) concentrations of odd-chain *n*-C₂₃ to *n*-C₃₁ alkanes, (c) *P*ₐq values, and (d) CPI values of lake and lakeshore sediments in Lake Wuliangsu.**

**3.2 Concentrations of *n*-alkanes**

Total concentrations of $n$-$C_{23}$ to $n$-$C_{31}$ alkanes in lake sediments show strong variations, ranging from 0.62 µg/g to 137.05 µg/g (Figure. 2b, Supplementary Table S1). The lowest concentrations were observed in sediments in the northern vegetated area (sites S1 and S2), whereas the highest concentration were in the open-water area (sites S3 and S11, Figure. 2b).



Considering that the total concentrations of $n$-C$_{23}$ to $n$-C$_{31}$ alkanes from submerged macrophytes and emergent plants are

132.75 μg/g (n=2) and 67.29 μg/g, respectively, sediments with $n$-alkane concentrations higher than 67.29 μg/g (i.e., sample S3) should be strongly contributed by submerged macrophytes instead of emergent plants. The total $n$-alkane concentration of the riverine sediment is 2.84 μg/g, much lower than those from submerged macrophytes and emergent plants. The total concentrations of $n$-C$_{23}$ to $n$-C$_{31}$ alkanes from the lakeshore sediments range from 0.79 μg/g to 3.54 μg/g, which are close to the lower end of lake sediments and the riverine sediment. The highest concentrations in lakeshore sediments were observed

in the northeastern area (site SS2), whereas the lowest concentrations at the southeastern area (site SS5).

There was a significant positive correlation between TOC values and the concentrations of $n$-C$_{23}$ to $n$-C$_{31}$ alkanes among all sediments except for sample S3 ($r^2$=0.74, n=20, $p < 0.01$, Supplementary Figure S1). Such significant positive correlation also exists for all the individual alkane compound ($r^2$=0.65, 0.66, 0.71, 0.81, 0.85 for $n$-C$_{23}$, $n$-C$_{25}$, $n$-C$_{27}$, $n$-C$_{29}$, and $n$-C$_{31}$, respectively, n=20, $p < 0.01$). Considering that sample S3 contains a much higher $n$-alkane concentration, which is close to

those of submerged macrophytes, we suggested that this sample might be largely influenced by pure supply from submerged macrophytes, and thus showed its unique correlation with TOC. Therefore, $n$-alkanes are still representative for most of sedimentary OC in Lake Wuliangsu.

### 3.3 Distribution patterns of $n$-alkanes

$n$-Alkanes of aquatic plants and surface sediments range from $n$-C$_{17}$ to $n$-C$_{31}$, but show different distribution patterns in terms of mid- to long-chain compounds (Figure. 3a–c). For instance, submerged macrophytes were basically dominated by $n$-C$_{23}$ and $n$-C$_{25}$, whereas emergent plants show the predominance of $n$-C$_{27}$ and $n$-C$_{29}$ components and trace occurrence of $n$-C$_{31}$ alkane. Such $n$-alkane distribution patterns are similar to aquatic plants collected from lakes in Qinghai-Tibet Plateau (Aichner et al., 2010; Liu et al., 2015; Liu and Liu, 2016) and lower reaches of the Yangtze River (Yu et al., 2021). $n$-Alkanes of the

riverine sediment (sample WJ) show a predominance of the $n$-C$_{29}$ and $n$-C$_{31}$ alkanes with low odd-chain predominance, slightly different from those of the emergent plants (Figure. 3). Accordingly, the $P_{aq}$ values, the proxy evaluating the relative contributions of $n$-alkanes from aquatic plants (Ficken et al., 2000), are highest in submerged macrophyte (0.82), moderate in emergent plant (0.43) and lowest in riverine soil (0.21). The CPI values describing the odd-over-even predominance (Eglinton and Hamilton, 1967) are also highest in submerged macrophytes (18.02), moderate in emergent plants (10.24), and lowest in

riverine soil (6.03, Figure. 2).

The distribution pattern of $n$-alkanes in lake sediments and lakeshore sediments show mixing signatures from submerged macrophytes, emergent plants, and riverine soil (Figure. 3d–f). Most of the sediments are characterized by $n$-C$_{29}$ alkane as main peak, except for samples collected from sites S3 and SS4 characterized by $n$-C$_{25}$ alkane as main peak. In lake sediments, the $P_{aq}$ values range from 0.21 to 0.51, with higher values in samples located at both the northern (site S3) and the southern

(site S13) open-water areas and lower values in samples located at the lake center (site S6, Figure. 2c). The CPI values range



from 5.79 to 19.18, and show a strong correlation with the $P_{aq}$ values ($r^2$=0.84, n=14, $p < 0.01$). The relatively high CPI values were observed at the southern (sites S11, S12, and S13) and the northern (site S3) open-water areas, whereas the lowest values were roughly at the lake center (site 10, Figure. 2d). In terms of lakeshore sediments, sample SS4 is characterized by relatively higher concentrations of $n$-$C_{24}$ alkane, highest $P_{aq}$ value (0.51) and lower CPI value (2.83). Together with the carbon isotopic

evidence (Supplementary Table S1), we suspected that this sample might be contaminated by artificial products containing $n$-$C_{23}$, $n$-$C_{24}$, and $n$-$C_{25}$ alkanes, and thus excluded this sample from any further discussion and evaluation. For the rest of the lakeshore sediments, the $P_{aq}$ values and the CPI values range from 0.25 to 0.47 and from 3.62 to 7.26, respectively. The highest $P_{aq}$ value was observed in the southern open-water area close to outlet channel (0.47, site SS7) and the lowest value was in the southeast far from the open-water area (0.25, site SS6). The highest CPI value was observed in the southeast far from the open-

water area (7.26, site SS5), and the lowest value was close to the main inlet channel (3.62, site SS3). Different from the lake surface sediments, the CPI values of lakeshore sediments show weak but negative correlation with the $P_{aq}$ values ($r^2$=0.10, n=6), tentatively suggesting that the even-chain $n$-alkanes of lakeshore sediments might be contaminated by oil contamination to some extent.

**3.4 $\delta^{13}C$ signatures of $n$-alkanes**

The $\delta^{13}C$ values of mid-and long-chain odd $n$-alkanes ($n$-$C_{23}$, $n$-$C_{25}$, $n$-$C_{27}$, $n$-$C_{29}$, and $n$-$C_{31}$) ranged from –36.6‰ to –24.1‰ (Figure. 3). Submerged macrophytes exhibited the most $^{13}C$-enriched $n$-alkanes (–26.2‰ to –24.1‰), while $n$-alkanes from emergent plants were more depleted in $^{13}C$ values (–36.6‰ to –34.0‰), $n$-alkanes in riverine sediment show intermediate values (–33.1‰ to –30.7‰). The $\delta^{13}C$ values for mid- and long-chain $n$-alkanes of lake sediments range from –34.84‰ to –

23.98‰ (Supplementary Table S1). In all sediments but sample S1, $\delta^{13}C$ values of $n$-alkanes show decreasing trend along with carbon length from $n$-$C_{25}$ to $n$-$C_{31}$. $\delta^{13}C$ values of both mid- and long-chain $n$-alkanes in sediments at the open-water area (sites S3, S11, S12, S13, and S14) are more positive than those in other samples (Supplementary Table S1). In terms of lakeshore sediments, $\delta^{13}C$ values for mid- and long-chain $n$-alkanes ranged from –36.5‰ to –26.0‰. $\delta^{13}C$ values of $n$-alkanes show decreasing trend along with carbon length in samples SS1, SS3, SS5, and SS6. While $n$-$C_{23}$ and $n$-$C_{25}$ in SS4 were relatively

depleted in $^{13}C$ (–34.4% to –34.3%), further supplementary the contamination by artificial products. The $\delta^{13}C$ values for $n$-$C_{31}$ were the most positive in samples SS2 and SS7, tentatively suggesting more contribution of submerged macrophytes in single carbon at two sites.



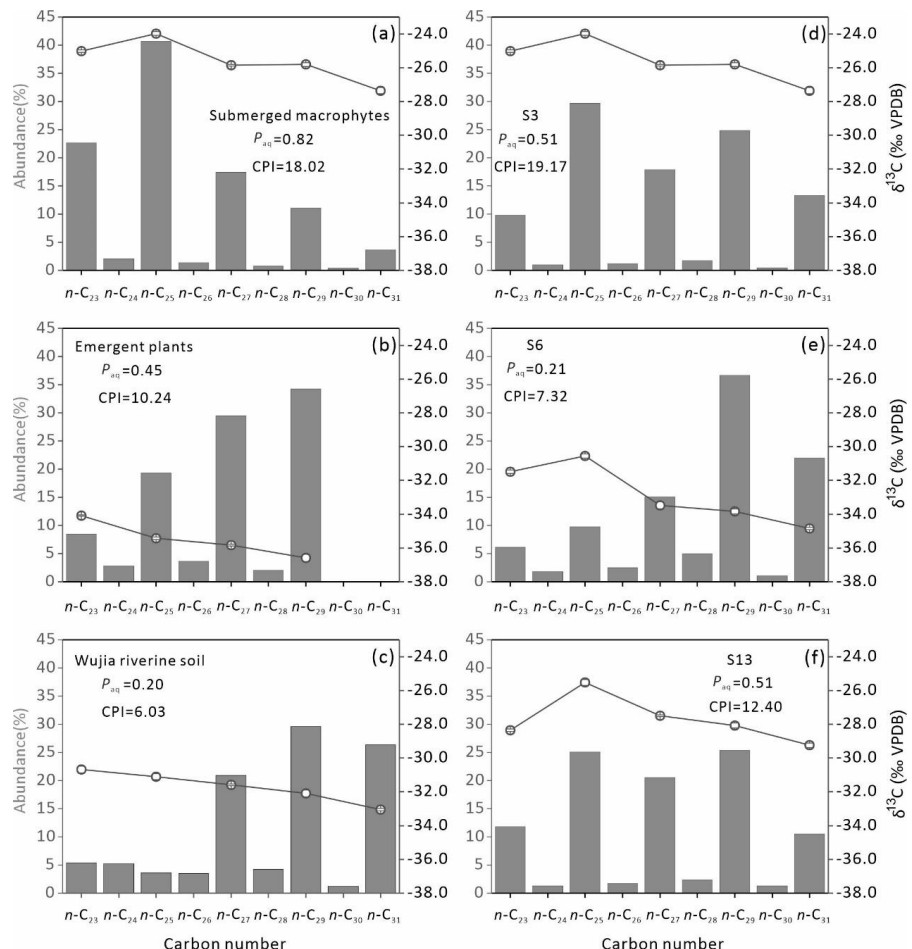

**Figure 3. Bar plots and corresponding line graphs showing the distribution of the mid- and long-chain *n*-alkanes (relative abundance,**

**left axis) and their stable carbon isotopic composition (right axis), for six representative sediments from the Lake Wuliangsu,**

**including (a) submerged macrophytes (b) emergent plants (c) riverine soil (d) S3 (e) S6 (f) S13.**

## *4. Discussion*

### 4.1 Different *n*-alkane characteristics among various OC sources of Lake Wuliangsu

In Lake Wuliangsu, allochthonous and autochthonous OC showed distinct distribution patterns and $\delta^{13}C$ values of mid-

and long-chain *n*-alkanes (Figure. 3). Generally, submerged macrophytes were characterized by a predominance of *n*-$C_{23}$ and

*n*-$C_{25}$ alkanes and low concentrations of even-numbered chain *n*-alkanes, resulting in higher $P_{aq}$ (0.82) and CPI (18.02) values.

The $\delta^{13}C$ values of *n*-alkanes in submerged macrophytes were more $^{13}C$-enriched (–24.1‰ to –26.2‰), consistent with other

studies on shallow lakes globally (e.g., Aichner et al., 2010; Liu et al., 2015; Liu and Liu, 2016; Yu et al., 2021). Similar to

emergent plants collected in many Chinese shallow lakes (e.g., Lake Shijiu, Lake Yangcheng, Lake Xihu, and Lake Hong,





Duan and He, 2011; Ho et al., 2015; Yu et al., 2021), emergent plants from Lake Wuliangsu exhibited a predominance of $n$-$C_{27}$ and $n$-$C_{29}$ alkanes, low concentrations of even-numbered chain $n$-alkanes, and under-detectable $n$-$C_{31}$ alkane, resulting in lower $P_{aq}$ (0.43) and higher CPI (10.24) values. The $n$-alkanes of emergent plants were more depleted in $^{13}$C (–34.0‰ to –36.6‰). The up to ~10‰ difference of $\delta^{13}$C values of $n$-alkanes between submerged macrophytes and emergent plants might

be attributed to their different photosynthesis pathways, considering that submerged macrophytes use $HCO_3^-$ as carbon source for photosynthesis, whereas emergent plants use $CO_2$ as the primary carbon source (Chappuis et al., 2017). The riverine sediments were characterized by a higher abundance of long-chain $n$-alkane homologues (i.e., $n$-$C_{29}$ and $n$-$C_{31}$), resulting in a much lower $P_{aq}$ value (0.20). Moreover, the relative concentrations of even-numbered $n$-alkanes in riverine sourced sediments were relatively higher than those from both aquatic plants, resulting in a slightly lower CPI value (6.03). The $\delta^{13}$C values of $n$-

alkanes from the riverine sediment range from –30.7‰ to –33.1‰, suggesting that the riverine sourced OC might be mainly derived from $C_3$ plants (Meyers, 2006).

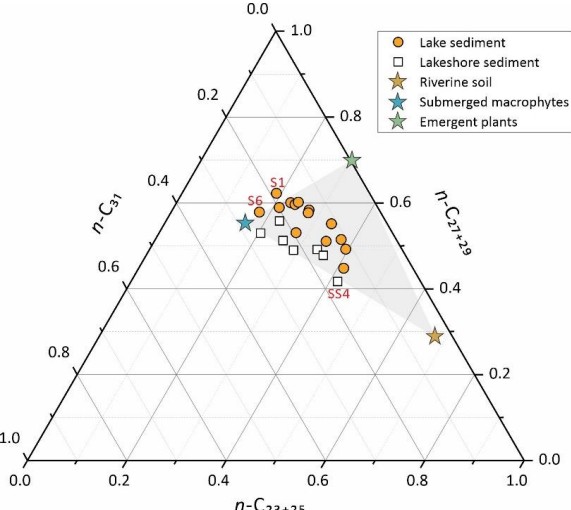

**Figure 4. Ternary diagrams of relative percentage of $n$-$C_{23}$+$n$-$C_{25}$, $n$-$C_{27}$+$n$-$C_{29}$, and $n$-$C_{31}$ relative to $n$-$C_{23+25+27+29+31}$ for all samples including lake surface sediments (orange dots), lakeshore sediments (white squares),**

**riverine soil and aquatic plants (stars).**

The notion of quantitative/semi-quantitative methods based on distribution patterns and $\delta^{13}$C values of $n$-alkane was further supported by the results of lake sediment samples. Firstly, on the ternary diagram of the relative abundance of $n$-$C_{23+25}$, $n$-$C_{27+29}$, $n$-$C_{31}$ to $n$-$C_{23+25+27+29+31}$ alkanes (%$C_{23+25}$, %$C_{27+29}$, and %$C_{31}$), most lake sediments roughly fall into the area

delineated by the three end-members of submerged macrophytes, emergent plants, and riverine soil (Figure. 4). Secondly, all the $\delta^{13}$C values of mid- and long-chain $n$-alkanes from surface sediments also roughly fall into the $\delta^{13}$C range among these





three end-members (–24.0‰ to –36.7‰, supplementary Table S1). Thirdly, more positive δ$^{13}$C values were observed in $n$-C$_{23}$ and $n$-C$_{25}$ homologues than $n$-C$_{29}$ and $n$-C$_{31}$ homologues for all sediment samples, tentatively suggesting a major contribution of mid-chain $n$-alkanes from submerged macrophytes as well as contributions from emergent plants and terrestrial plants in

long-chain $n$-alkanes (Table S1). These pieces of evidence together demonstrate mixture sources of surface sedimentary OC from submerged macrophytes, emergent plants, and riverine materials, which were also widely recognized by previous studies from bulk characteristics of OC (i.e., Sun et al., 2006; Tian et al., 2011; Fu et al., 2013; Geng et al., 2021). Therefore, the distribution pattern and the δ$^{13}$C values of mid- and long-chain $n$-alkanes showed great potential to semi-quantitatively/quantitatively differentiate contribution among various sedimentary OC sources in Lake Wuliangsu.


**4.2 Quantification on allochthonous and autochthonous OC by end-member mixing models**

**4.2.1 Model #1: solely based on distribution patterns of n-alkanes**

Firstly, we developed a three end-member model solely based on distribution pattern (i.e., %$n$-C$_{23+25}$, %$n$-C$_{27+29,}$ and %$n$-C$_{31}$, Figure 5), to quantitatively estimate the relative contributions of allochthonous and autochthonous OC sources to the

sediments within Lake Wuliangsu. Basically, the relative contributions of OC from submerged macrophytes, emergent plants, and riverine sources ($F_{sub}$, $F_{emer}$, and $F_{riv}$) can be calculated by Eq. (3)–(6) listed in Supplementary Material. Sample S1, S6, and SS6 fall slightly outside the area circled by the three sources, we assigned $F_{sub}$ values of samples S1 and S6 to 0, whereas $F_{emer}$ value of sample SS6 to 0.

The results of Model #1 show that the relative contributions of mid- and long-chain $n$-alkanes from submerged

macrophytes, emergent plants, and riverine sources to lake surface sediments were 15.2% (0%–42.9%), 31.8% (14.3%–42.6%), and 53.0% (34.5%–68.6%), respectively. The relative contributions of mid- and long-chain $n$-alkanes from submerged macrophyte, emergent plant, and riverine soil to lakeshore sediments were 20.1% (8.4%–32.7%), 9.4% (0%–18.6%), and 70.5% (53.1%–91.0%), respectively. The major contributions of riverine sourced OC to the $n$-alkane pool in both lake sediments and lake shore sediments seem to contradict with the conclusion that sedimentary OC is mainly of autochthonous origins within

Lake Wuliangsu (Sun et al., 2006; Tian et al., 2011; Fu et al., 2013).

As the CPI values and δ$^{13}$C values of $n$-alkanes are independent with the distribution pattern of $n$-C$_{23}$ to $n$-C$_{31}$ alkanes, they could be applied to test whether this model was valid or not. We calculated the estimated CPI (CPI$_{est}$) values and δ$^{13}$C values of every single $n$-alkane component (δ$^{13}$C$_{i\text{-est}}$) based on the results of this mixing Model #1, by Eq. (7) and (8) listed in Supplementary Material. On one hand, the CPI$_{est}$ values for the lake sediments and lakeshore sediments were linearly correlated

with the CPI values ($r^2$=0.45, n=20, $p$ < 0.01, Figure. 5a), but with slightly different from the absolute values (difference of –4.60 to 9.85). Specially, the samples with lower CPI$_{est}$ values than the actual CPI values are mainly from open-water areas (e.g., sites S3, S11, S12, S13, and S14), tentatively indicating that this mixing model might underestimate the contribution of submerged macrophyte OC at these sites.



Although the CPI$_{est}$ values for lake sediments also show positive correlation with actual CPI values ($r^2$=0.59, n=14,

$p$<0.01), those for lakeshore sediments show very weak but negative correlation with actual CPI values ($r^2$=0.16, $n$=6).

Therefore, the end-member mixing model solely based on distribution patterns of $n$-alkanes might not be applicable for the

lakeshore areas, considering that the relative concentration of odd and even-chain $n$-alkanes have altered caused by complex

microbial degradation behavior and OC sources. On the other hand, the estimated $\delta^{13}$C values of every single $n$-alkane

exhibited a significant correlation with the actual $\delta^{13}$C values ($r^2$= 0.36, n=100, $p$ < 0.01, Figure. 5b), but also showed slightly

different absolute values (difference of -4.1‰ to 6.0‰). The data biasing the 1:1 line for the estimated and the actual $\delta^{13}$C

values with 2‰ uncertainty is mainly from $\delta^{13}$C values of submerged macrophyte-sourced $n$-alkanes (i.e., $n$-C$_{25}$) from sites

S1, S3–S5, S7, S9 and S11–S14, and $\delta^{13}$C values of $n$-C$_{27}$, $n$-C$_{29}$ and $n$-C$_{31}$ alkanes from lake sediments mainly at the open-

water areas (sites S3, S11, S12, S13 and S14, Figure. 5b). Accordingly, results from Model #1 seem to underestimate the

relative contributions from submerged macrophytes, consistent with what was found from the comparison of CPI$_{est}$ and actual

CPI values.

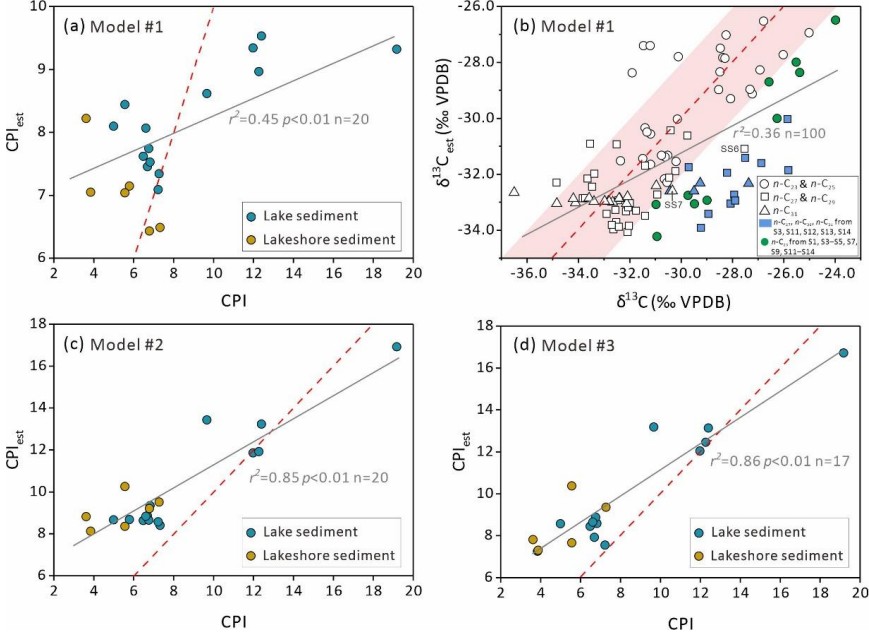

**Figure 5.** Plots of (a) actual CPI values *vs.* CPI$_{est}$ values calculated by Model #1, (b) actual $\delta^{13}$C values *vs.* estimated $\delta^{13}$C values of

$n$-C$_{23}$ to $n$-C$_{31}$ calculated by Model #1 (blue colored refer to $n$-C$_{27}$, $n$-C$_{29}$ and $n$-C$_{31}$ from open water area, e.g. S3, S11, S12, S13, and

S14, and green colored refer to $n$-C$_{25}$ from S1, S3-S7, S9, and S11-S14), (c) actual CPI values *vs.* CPI$_{est}$ values calculated by Model

#2, and (d) actual CPI values *vs.* CPI$_{est}$ values calculated by Model #3, respectively from sediments profile of Lake Wuliangsu. The

red dashed line represents 1:1 correlation between CPI$_{est}$ values and actual CPI vales. The linear regression line is shown in gray.

In summary, Model #1 solely based on the distribution pattern of $n$-alkanes can only track the spatial variation of the



relative contributions of submerged macrophytes, emergent plants, and riverine sources within the lake rather than the

surrounding catchment. Even though, there might be large uncertainty in the absolute value from this model. The potential

reason might be the insufficient samples for the end-member determination and more samples from aquatic species and riverine

sources are needed. Another reason is that the microbially mediated degradation on shorter chain *n*-alkanes during the

transportation and depositional processes, which to some extent resulting in relatively higher contribution from *n*-alkanes of

longer chains (i.e., signature from emergent plants and riverine sourced OC, Van Beilen et al., 2003; Wang et al., 2019). Both

difficulties might be constrained by the introduction of the $\delta^{13}$C values of *n*-alkanes.

**4.2.2 Model #2: introducing the $\delta^{13}$C values of n-alkanes into the mixing model**

We further developed another end-member mixing model for lake surface sediments (Model #2) together with the $\delta^{13}$C

values of *n*-$C_{23}$ to *n*-$C_{31}$ alkanes. For the $F_{sub}$, we first listed all the possible cases with $f_{emer}/f_{riv}$ ranging from 0 to infinity, and

calculated the averaged $F_{sub}$ value for all possible $F_{sub}$ values following Eq. (9)–(11) listed in Supplementary Material. Similarly,

for the $F_{emer}$, we first listed all the possible cases with $f_{sub}/f_{riv}$ ranging from 0 to infinity, and calculated the averaged $F_{emer}$ value

for all possible $F_{sub}$ values $f_{emer}$ following Eq. (12)–(14) listed in Supplementary Material, and then $F_{riv}$ was calculated by Eq.

(15).

According to Model #2, relative contributions of mid- and long-chain *n*-alkanes from submerged macrophytes, emergent

plants, and riverine sources in lake surface sediments were 42.1% (14.0%–95.5%), 26.1% (1.9%–43.4%), and 31.8% (2.6%–

44.5%), respectively (Supplementary Table S3). The relative contributions of mid- and long-chain *n*-alkanes from submerged

macrophytes, emergent plants, and riverine sources to lakeshore sediments were 29.3% (12.9%–48.5%), 30.8% (13.8%–

50.6%), and 40.0% (30.9%–47.9%), respectively (Supplementary Table S3). Compared to the results from Model #1, the ones

from Model #2 showed lower riverine OC contribution (avg. 34.3% *vs.* 58.3%), slightly higher emergent plant contribution

(avg. 27.5% *vs.* 25.1%), and much higher submerged macrophyte contribution (avg. 38.2% *vs.* 16.7%). Besides that, the spatial

variation patterns of the relative contributions from different OC sources are quite similar, as illustrated by strong correlations

of the same proxy between both models ($r^2$=0.61 and 0.50 for $F_{sub}$ and $F_{riv}$, n=20, *p*<0.01). Specifically, the contribution of

submerged macrophytes was relatively higher at sites S3, S11, S12, S13, S14, and SS7 in Model #2, in agreement with the

results from Model #1. Therefore, both independent models might be capable to track the spatial characteristics of

allochthonous and autochthonous OC contributions to the Lake Wuliangsu sediments but show different absolute values for

both submerged macrophytes and riverine soils.

We also calculated the CPI$_{est}$ values based on the results of Model #2 to test whether this model was valid or not

(Supplementary Table S3). The CPI$_{est}$ values more significantly correlated with the actual CPI values in Model #2 than Model

#1 ($r^2$=0.85, n=20, *p*<0.01, Figure. 5c). The CPI$_{est}$ values from lakeshore sediments in Model #2 also show weak positive

correlation with actual CPI values ($r^2$=0.26, n=6), much more reasonable than those from Model #1 (Figure. 5a). The absolute





CPI$_{est}$ values from Model #2 are much closer to the real CPI values than those from Model #1 (differences of –4.71 to 2.26 vs. –4.60 to 9.85), suggesting that Model #2 might be more convincing in terms of the absolute values. Indeed, between the two models, higher contributions from submerged macrophytes occurred in samples located in the open-water area (sites S3, S11, S12, S13, and S14) inferred by Model #2 (62.2%–95.5%) should be more rational. Especially for sample S3, results from

Model #2 demonstrated that sedimentary $n$-alkanes were almost exclusively contributed by submerged macrophytes (95.5%), potentially explaining why this sample deviating the linear correlation between the total concentration of $n$-alkanes and TOC value by its much higher $n$-alkanes concentrations (Figure. 2b). In this sense, introducing the $\delta^{13}$C signatures into the models of $n$-alkanes to reduce the potentially large error brought from uncertainties of the distribution patterns in the sourced appointment.


### 4.2.3 Model #3: combination of both Model #2 and Model #3

    In Model #2, we initially listed all possibilities of $f_{emer}/f_{riv}$ ranging from 0 to infinity, and calculated the average value of $F_{sub}$, $F_{emer}$, and $F_{riv}$. In this way, uncertainties would be produced, as possibilities of different $f_{emer}/f_{riv}$ are not weighted by other methods. In this sense, we attempted to fixed $f_{emer}/f_{riv}$ to the $F_{emer}/F_{riv}$ values that calculated from Model #1, and calculate $F_{sub}$

based on Eq. (9)–(11), and then calculate $F_{emer}$ and $F_{riv}$ based on Eq. (16)–(17) listed in Supplementary Material. In this model we excluded samples S1, S6, and SS6 from the end-member mixing model to avoid potentially improper estimation, considering their results derived from Model #1 are partially unreliable.

    The results from this updated Model #3 show that the relative contributions of mid- and long-chain $n$-alkanes from submerged macrophytes, emergent plants, and riverine sources in lake surface sediments were 43.9% (13.0%–95.3%), 22.4%

(1.2%–37.2%), and 33.7% (3.6%–61.3%), respectively (Supplementary Table S3). The relative contributions of mid- and long-chain $n$-alkanes from submerged macrophytes, emergent plants, and riverine sources to lakeshore sediments were 34.5% (20.8%–56.4%), 9.8% (2.0%–16.9%), and 55.6% (34.4%–73.5%), respectively (Supplementary Table S3). Results from this updated Model #3 are strongly correlated with those from Model #2 results ($r^2$=0.95 and 0.81 for submerged macrophytes and riverine OC, n=17, $p < 0.01$). Also, the CPI$_{est}$ values from Model #3 showed significantly correlated with those from Model

#2 ($r^2$=0.97, n=17, $p$<0.01), and are strongly correlated with the actual CPI values ($r^2$=0.86, n=17, $p$<0.01, Figure. 5d). The absolute CPI$_{est}$ values from Model #3 are also much closer to the actual CPI values than those from Model #1 (differences of –4.83 to 2.46 vs. –4.60 to 9.85). This means that once $\delta^{13}$C signatures of $n$-alkanes were introduced (i.e., Model #2 and Model #3), the end-member mixing models would be relatively stable no matter how we calculate the $f_{emer}$, $f_{sub}$, and $f_{riv}$ values. Therefore, all three models might be all adequate for the evaluation of spatial variation patterns of different OC sources of

Lake Wuliangsu, and Models #2 and Model #3 would be more convincing with respect to discuss the absolute values.



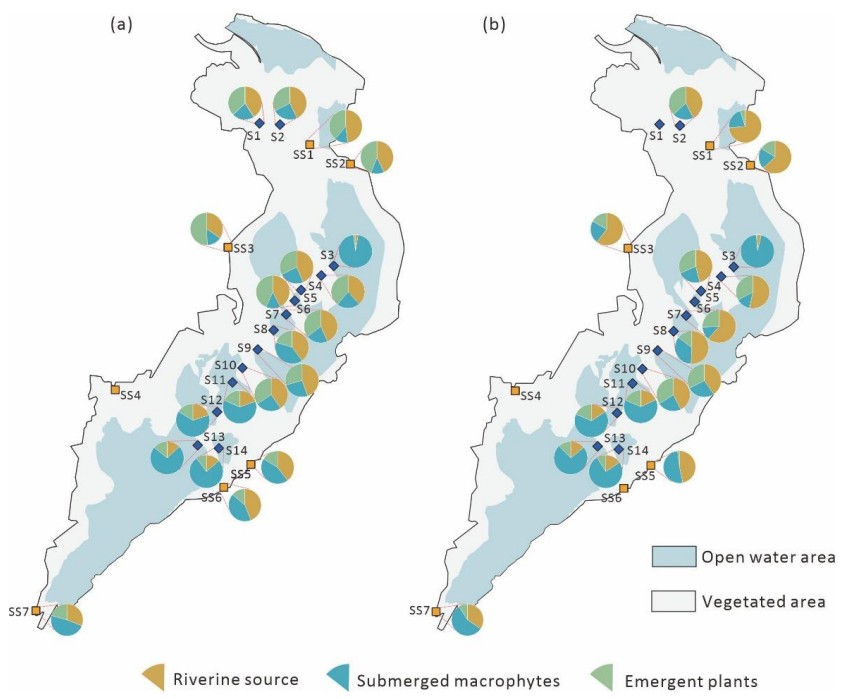

**Figure 6. Pie charts of OC contributions from riverine soil (yellow segment), submerged macrophytes (blue segment), and emergent plants (light-green segment) for lake and lakeshore surface sediments from Lake Wuliangsu calculated by (a) Model #2 and (b) Model #3.**


### 4.3 Spatial characteristics of allochthonous and autochthonous OC within Lake Wuliangsu

According to the results of both Model #2 and Model #3, the relative contribution of riverine sourced OC to sedimentary *n*-alkanes demonstrated a clear decreasing trend from the lake center (S7 and S8, 11.6%–12.4% from Model #2 and 18.5%–21.0% from Model #3) to the southern part (S13 and S14, 3.6%–4.0% from Model #2 and 4.0%–4.6% from Model #3, Figure.

6). In terms of the absolute contributions to sedimentary OC calculated by Eq. (18)–(20) listed in Supplementary Material, OC derived from riverine input was higher at sites S7 and S8 (0.57%–0.60% from Model #2 and 0.85%–1.09% from Model #3, Figure. 7), and exhibited a decreasing trend to the southward until it reaches the lowest values at site S14 (0.06% from Model #2 and Model #3). Therefore, a large amount of riverine sourced OC from the main channel has settled down during the southward migration process (Fu et al., 2013). This notion was also supported by the gradual transition of the sedimentary

grain size from sandy silt near the main inlet channel to fine grain at the off-shore areas of Lake Wuliangsu (Yang et al., 2009). On the one hand, once transported to Lake Wuliangsu, the majority of riverine OC can be quickly deposited in regions close to the main inlet channel by hydrological sorting due to quick diminished of the hydrodynamics (near site S6, Zhang et al., 2010), with minor riverine OC further transporting to the southern outlet of Lake Wuliangsu. On the other hand, emergent plant belt at the vegetated areas would also block the coarse particles from further transportation due to its robust underground



rhizome system (Zhao et al., 2012). Interestingly, the riverine sourced OC at samples located at the southern open-water of

Lake Wuliangsu (sites S12–S14, riverine sourced OC of 0.06%–0.41%, TOC of 4.45%–7.19%) are very low and are close to

those from lake sediments at the northern section (sites S1 and S2, riverine sourced OC of 0.10%–0.20%, TOC of 0.64%–

0.88%) and lakeshore sediments surrounding the lake (sites SS1–SS7, riverine sourced OC of 0.02%–0.31%, TOC of 0.15%–

1.30%, Figure. 7). Considering that lake sediments at the northern section and lakeshore sediments surrounding the lake are

characteristic by relatively stronger hydrodynamics, lower TOC values, and less impact from the riverine input from the

channels in the western main inlet channel, most of the riverine sourced OC from the western main inlet channel have been

deposited within Lake Wuliangsu, at least before reaching to sites S12–S14. Therefore, lake Wuliangsu may be act as an

important trap and sink for allochthonous OC. The soil materials transported to the Middle and the Lower Yellow River

Reaches from Lake Wuliangsu might be mainly from the catchment instead of the Upper Yellow River Reaches.

For the autochthonous OC, they contribute higher to sedimentary $n$-alkanes than riverine sourced OC for all samples (avg.

65.7% from Model #2 and 59.9% from Model #3). For sites located in the open-water area (sites S3, S11, S12, S13, and S14),

autochthonous OC show predominated contributions to sedimentary $n$-alkanes (80.0%–97.4% from Model #2 and 81.9%–

96.5% from Model #3), with a major contribution from submerged macrophytes (62.2%–95.5% from Model #2 and 63.4%–

95.3% from Model #3) and minor from emergent plants (1.5%–18.2% from Model #2 and 1.2%–18.5% from Model #3). For

the rest of the lake surface sediments, autochthonous OC are still the main contributors for sedimentary $n$-alkanes of Lake

Wuliangsu (55.5%–61.5% from Model #2 and 47.1%–59.5% from Model #3), with roughly comparable contribution from

both submerged macrophytes (14.0%–39.8% from Model #2 and 13.7%–34.6% from Model #3) and emergent plants (20.3%–

43.4% from Model #2 and 14.7%–37.2% from Model #3). In lakeshore sediments, contributions from autochthonous OC to

sedimentary $n$-alkanes are similar to the lake surface sediments outside the open-water area (52.1%–69.1% from Model #2

and 26.5%–65.6% from Model #3).

In terms of the absolute contributions to sedimentary OC (Figure. 7), the contribution of both submerged macrophyte-

derived OC and emergent plant-derived OC are strongly coupled with the bulk TOC values (e.g., $r^2$=0.71 and 0.62, n=20 from

Model #2, and $r^2$=0.72 and 0.66, n=17 from Model #3, $p$<0.01), further supplementary the notion that the sedimentary OC of

Lake Wuliangsu is mainly of the autochthonous source. Specifically, submerged macrophytes contribute much higher in the

open-water areas from northern part (site S3, 3.71% from Model #2 and 3.76% from Model #3) and southern part (sites S11,

S12, and S13, 3.04%–4.39% from Model #2 and 3.13%–4.30% from Model #3). The contribution of submerged macrophytes-

derived OC showed an increasing trend from the central section (site S4) to the southern part (sites S11–S13, Figure. 7),

consistent with the spatial distribution pattern of submerged macrophytes as inferred by the $P_{aq}$ values (Figure. 2) from the

same samples and the results from the on-site investigations (Du et al., 2022). For sites along the central to southern section

(sites S6–S13), contributions of the emergent plant-derived OC are higher (1.16%–3.61% from Model #2 and 1.14%–3.33% from

Model #3). Therefore, higher contributions of submerged macrophytes are usually observed in the open-water area, while



higher contributions of emergent plants in the central part and southern part of lake across both the vegetated area and the
open-water area. Such results are consistent with the remote sensing data and on-site investigations (Yu et al., 2007; Bao et al.,
2016), which suggest that major contributions of emergent plants to OC deposition and stronger difference among areas with

different water depth and distances to the lake shore.

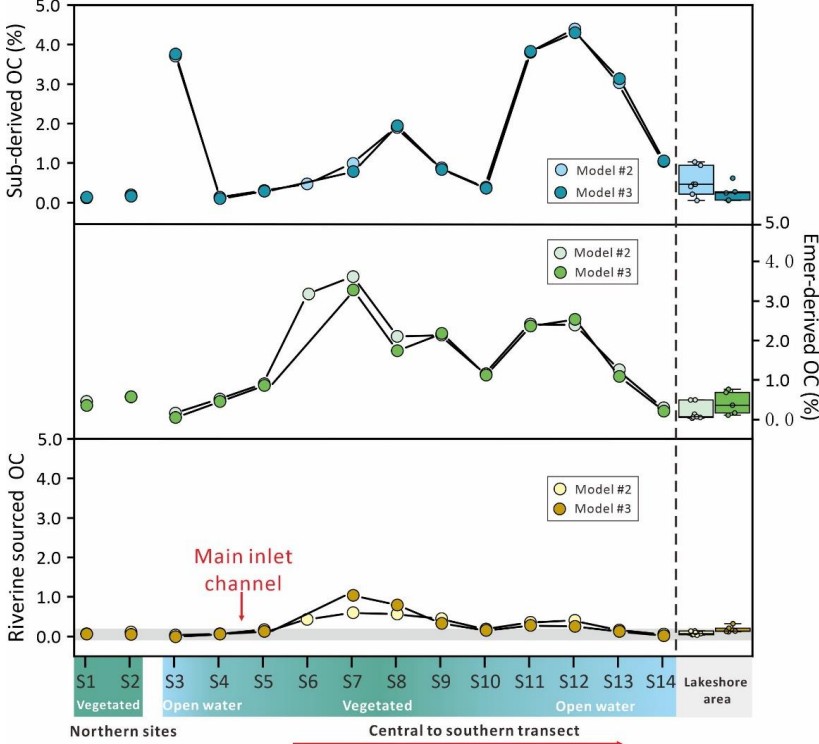

**Figure 7. Absolute OC contributions from submerged macrophytes, emergent plants, and riverine soil calculated by Model #2 and Model #3. For lake shore samples, data were shown in box plot. The boxes, the horizontal lines within boxes, and the whiskers indicate the interquartile range, median values, and minimum and maximum values, respectively.**


Theoretically, water transparency, water depth, and nutrient concentrations are the key parameters affecting the growth
of submerged macrophytes and emergent plants. With stable hydrological conditions and less anthropogenic disturbances in
the open-water areas (sites S3, S11, S12, and S13, Tian et al., 2020), higher light energy transmission in clear water body could
favor the growth of submerged macrophytes (Heidbuechel et al. 2019). In the case of Lake Wuliangsu, submerged macrophytes

grow quickly in the ice-free seasons, and experience massive death, corruption, and deposition during the frozen season,
eventually promoting strong OC burial (Huang et al., 2015). By contrast, low biomass of submerged macrophytes was observed
at regions close to main channel with low water transparency, largely due to the stronger sediment resuspension under intensive
hydrodynamics (Du et al., 2022). Interestingly, the open-water areas located at the central part (sites S11, S12, and S13) show



both relatively higher water transparency and water depth (Tian et al., 2020), we suspect that the water depth and offshore
distance also act an important role for water transparency. As emergent plants contribute similarly in the central and southern
sections of lake across both the vegetation-covered area and the open-water area, the water transparency and water depth might
not be the major factors for the growth of the emergent plants (Yu et al., 2007). Considering Lake Wuliangsu is generally
shallow (1–1.5 m) under current intense paludification, the whole lake might be all suitable for the emergent plants to grow.

Interestingly, at northern sites with relatively higher contribution of emergent plants over submerged macrophytes (sites
S1 and S2), the absolute TOC are lower, tentatively suggesting that the productions of emergent plants might be controlled by
factors from the central and the south. Therefore, we believe that higher concentrations of $NH_4^+$ and $PO_4^{3+}$ from the agricultural
farmland could trigger the bloom of submerged macrophytes and emergent plants (Ciurli et al., 2009), as inferred by higher
OC contributions from both submerged macrophytes and emergent plants at sites S11, S12, and S13 corresponding to higher
nitrogen and phosphorous concentrations. The decomposition of litters could continuously release the nutrients (Wu et al.,
2011), creating a positive feedback loop for the submerged macrophytes to nourish. However, much higher concentrations of
$NH_4^+$ and $PO_4^{3+}$ at this region might somehow hinder the submerged macrophytes and emergent plants from prospering (Liu
et al., 2020). Therefore, to improve the ecological conditions and OC burial of Lake Wuliangsu, it is necessary to supervise
and manage the quality of the inflowing water from the agricultural farmland, as well as preservation of the water body from
current intense paludification.


**4.4 Effect of OC sources to heterotrophic microbial activities in Lake Wuliangsu**

Among all OC mineralization processes, heterotrophic microbial attack on OC has been considered to be one of the most
critical factors in determining the sinking efficiency of OC in lakes (Mayer et al., 2006; Sun et al., 2022). Therefore,
investigations on heterotrophic microbial activities response to different types of OC sources would have important
implications in environmental management with respect to OC source-sink behaviors. A previous study suggested that
branched glycerol dialkyl glycerol tetraethers (brGDGTs) and GDGT-0 in Lake Wuliangsu should be mainly produced by
heterotrophic anaerobic bacteria (*Acidobacteria*) and methanogenic archaea (*Euryarchaeota*) within the lake, while
crenarchaeol from the soil OC transported through the Yellow River system (He et al., 2023). Accordingly, the BIT values of
lake surface sediments in Lake Wuliangsu are very close to 1.0 (He et al., 2023), while the BIT values in soil and riverine
samples collected in the Yellow River Reaches (0.81, Wu et al., 2014; Yang et al., 2014). $BR_{in\text{-}situ/soil}$ ratio calculated from the
BIT values can be used to quantitatively evaluate the activities of *in-situ* heterotrophic anaerobic bacteria (*Acidobacteria*) to
overall microorganisms (He et al., 2023). Also, a diagnostic method by GDGT-0/crenarchaeol values ($R_{0/5}$) can be used for
identification of the methanogenic *Euryarchaeota* ($R_{0/5} > 2$, Blaga et al., 2009; He et al., 2023).

When comparing our quantitative evaluation of the submerged macrophytes, emergent plants, and riverine sourced OC
values with the $R_{0/5}$ and $BR_{in\text{-}situ/soil}$ ratio, we observe a strong correlation between the submerged macrophyte-sourced OC and



both the $R_{0/5}$ and $BR_{in-situ/soil}$ ratios ($r^2$=0.68 and $r^2$=0.81, $p$<0.01, n=33, Figure. 8a–b, Supplementary Table S4), and no correlation between emergent plant-sourced OC $vs$. both the $R_{0/5}$ and $BR_{in-situ/soil}$ ratios ($r^2$=0.04 and $r^2$=0.10, n=33, Figure. 8c–d), and riverine sourced OC vs. both the $R_{0/5}$ and $BR_{in-situ/soil}$ ratios ($r^2$=0.02 and $r^2$=0.07, n=33, Figure. 8c–d). In addition, samples in the open-water zone usually show much higher $R_{0/5}$ and $BR_{in-situ/soil}$ ratios than vegetated zones (79.17 $vs$. 26.13 and

84.82 $vs$. 26.24 for $R_{0/5}$ and $BR_{in-situ/soil}$, respectively). These evidences suggest that the open water zones create ideal conditions for heterotrophic anaerobic bacteria and methanogenic $Euryarchaeota$ than the vegetated covered areas. Reasons for the phenomenon should be related to the geochemical and ecological properties of submerged macrophytes, and the environmental characteristics of open water zone. Firstly, compared with emergent plants and terrestrial plants, organic compounds sourced from submerged macrophytes are more labile and easier to be degraded in the water column and surface sediments (Bastviken

et al., 2003; Chimney and Pietro, 2006), creating a more anaerobic condition in places where the submerged macrophytes habitat (i.e. open water zone). Secondly, dense stands of submerged macrophytes could also provide more suitable substrates for methanogenic archaea to produce $CH_4$ (Hilt et al., 2022), as evidenced by higher $CH_4$ fluxes in the open water zones than vegetated areas of Lake Wuliangsu (Li et al., 2022). Such a notion is also supported by the incubation experiments showing that submerged macrophytes had higher $CH_4$ production than terrestrial debris in an anoxic sedimentary system (Grasset et al.,

2018). Thirdly, considering that the open-water zone usually occurs in the offshore region with relatively higher water depth (Li et al., 2022), we believe that the higher water depth and distance to the lake shore would both potentially promote the development of anoxia conditions for methanogenic $Euryarchaeota$ and anaerobic bacteria. Fourthly, when Lake Wuliangsu was skimmed with ice in the winter season, massive dead submerged macrophyte debris was deposited in the lake bottom, further consuming the dissolved oxygen availability in water column, which can be hardly recharged due to the frozen situation

on the lake surface (Sun et al., 2022; Zhang et al., 2022). All these reasons together lead to higher activities of heterotrophic anaerobic bacteria and methanogenic $Euryarchaeota$ in the open water zone observed in our data.

Interestingly, when the samples located at the open-water zone with predominated submerged macrophyte contributions are excluded (sites S3, S11, S12, S13, and S14), strong correlations can be also observed for the contributions of the riverine sourced OC $vs$. both the $R_{0/5}$ and $BR_{in-situ/soil}$ ratios ($r^2$=0.78 and $r^2$=0.82, respectively, $p$<0.01, n=25) and the emergent plant-

sourced OC vs. both the $R_{0/5}$ and $BR_{in-situ/soil}$ ratios ($r^2$=0.81 and $r^2$=0.74, respectively, $p$<0.01, n=25) (Figure. 8c–d). Indeed, results from a sedimentary core drilled in Lake Wuliangsu also show strong positive correlations for both the TOC values and the $R_{0/5}$ values during AD 1831–1972 and the TOC values and the $BR_{in-situ/soil}$ values AD 1831–2007 (He et al., 2023). In this sense, regardless of OC types, higher OC input are prone to consume the dissolved oxygen in the water body, and eventually create anaerobic conditions in freshwater systems for heterotrophic anaerobic microorganisms to prosper (Grasset et al., 2018).

However, as mentioned above, the submerged macrophytes conduct a much stronger effect on the heterotrophic anaerobic microbial activities. Therefore, the amount and more significantly the sources of OC can shape the microbial communities and reactivities, which would also play important roles in organic carbon cycle in shallow lake systems.

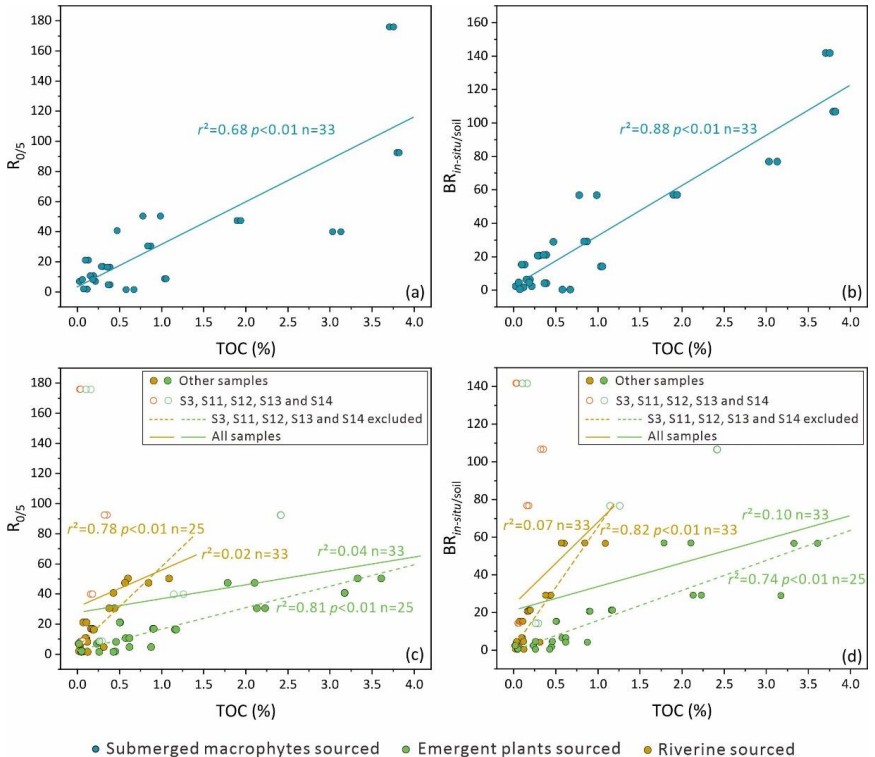

**Figure 8. Plots of (a) OC values derived from submerged macrophytes** *vs.* **$R_{0/5}$ values, (b) OC values derived from submerged macrophytes** *vs.* **$BR_{in-situ/soil}$ values, (c) OC values derived from emergent plants and riverine soil** *vs.* **$R_{0/5}$ values (d) OC values derived from OC values derived from emergent plants and riverine soil** *vs.* **$BR_{in-situ/soil}$ values . The linear regression line including all samples shown in solid line. The linear regression line excluding S3, S11, S12, S13, and S14 shown in dashed line.**

### 5. Conclusions

Our analyze of the distribution pattern and $\delta^{13}C$ value of *n*-alkanes from lake surface sediments yielded new insights about the OC appointment of Lake Wuliangsu with anthropogenic-induced paludification conditions. The findings highlight the potential importance of quantifying contribution from different sources to the sedimentary OC pool.

1) Submerged macrophytes, emergent plants, and riverine soil show their unique distribution patterns and $\delta^{13}C$ signatures of *n*-alkanes. In this sense, end-member mixing models were developed to quantify OC sources within lake. Compared to the

model based on distribution patterns of *n*-alkanes, the $\delta^{13}C$ values could effectively reduce the potential uncertainty of the model.

2) The model results suggest that the amount of riverine sourced OC from the main channel to Lake Wuliangsu have been settled down during the southward migration process. Therefore, Lake Wuliangsu serves as an important trap and sink for allochthonous OC from the Upper Yellow River Reaches. The model results also show a predominant contribution from the





autochthonous OC to Lake Wuliangsu, with the open-water area dominated by submerged macrophytes and the rest of the

areas by emergent plants. The spatial distribution pattern of autochthonous OC might be mainly controlled by water

transparency, water depth, and nutrient concentrations.

3) Open-water areas dominated by submerged macrophytes might be more favorable for heterotrophic anaerobic bacteria

and methanogenic archaea, largely due to active recycling processes for the labile OC derived from submerged macrophytes.


**Author Contributions**

Y.H. and design research and is responsible for the project. Y.H. and A.Z. collected the samples. Q.Z. and Y.H. performed

laboratory analyses. Q.Z. and Y.H. analyzed the data. Q.Z. and Y.H. wrote the initial draft of the manuscript. All authors edited

subsequent versions of the manuscript and approved the final submission.


**Acknowledgments**

We thank Prof. Yongge Sun for constructive suggestions on writing this paper. This study was supported by National Natural

Science Foundation of China (41877332, 42073071).

**Competing Interests**

The authors declare that they have no conflict of interest.

**Data availability**

The data that support the findings of this study are presented in the supplementary Table.



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
