# Peer review of "Quantification of allochthonous and autochthonous organic carbon in large and shallow Lake Wuliangsu based on distribution patterns and $\delta^{13}$ C signatures of *n*-alkanes"

_Biogeosciences, 2023_

## Author Comment (AC1)

Reply: We thank reviewer #1 for the valuable comments and suggestions on improving the manuscript. We have carefully followed the reviewer's suggestions revising our manuscript (see detailed responses below). We hope our response and the revised manuscript can be a satisfaction for the reviewer. FYI, we quoted line numbers regarding to the preprint manuscript online, and the revision will be updated once the reviewing process was completed.

**General comments**

This paper tried to use n-alkane proxies to reconstruct the contribution of different sources to sedimentary OM, which is significant for studying carbon cycling and carbon burial in lakes. The results from these proxies are generally credible and consistent with the current lake ecosystem systems and the natural background of the lake. This paper was well organized and written. I mainly suggested a further comparison of the results with those deduced from gross OM proxies (mainly gross OM carbon isotope).

Reply: Compared to *n*-alkanes, the gross OC can be more easily modified by degradation process during and after deposition (Mazeas et al., 2002; Jaffé et al., 2001). In this sense, the carbon isotope of gross OM might not be as indicative as the compound specific $\delta^{13}C$ values of *n*-alkanes (Ahad et al., 2011; Hyun et al., 2017). Comparison of $\delta^{13}C$ values between gross OM and *n*-alkanes would make another story of selective degradation of certain biomass or certain compounds, which is beyond the discussion of this paper focusing on the original sources for the organic carbon before degradation. Thus, we decided not to discuss gross OM carbon isotope in this manuscript and hope the reviewer can understand that.

**Major changes**

Line 30, the shallow lakes are generally defined as the mixing depth is larger than the maximal depth.

Reply: Accepted and revised this sentence (lines 31-32) as follows:

*"Shallow lakes (mixing depth exceeding the maximum depth, Qin et al., 2020), accounting for the largest area of lakes globally (Downing et al. 2006; Verpoorter et al., 2014)."*

Line 45, will phytoplankton contribute to the in situ productivity?

Reply: Yes, but phytoplankton only contribute a few to the total *in-situ* productivity. According to studies from Yu et al. (2007) and Ni et al. (2022), as well as our field investigations in recent years, Lake Wuliangsu is mainly covered by emergent plants (e.g., *Phragmites australis* and *Typha latifolia*) and submerged macrophytes (e.g., *Potamogeton pectinatus* and *Myriophyllum* spicatum). The amount of phytoplankton is relatively much lower. We have mentioned this situation in the manuscript (lines 63-64):

*"Alternatively, the amount of phytoplankton is relatively lower compared to the widespread emergent plants and submerged macrophytes (Ni et al., 2022)."*

Line 50, Zhu et al., 2018, Li et al., 2019, He et al., 2023. These are not original references about the function and carbon cycling associated with submerged macrophytes, please also adding the initial references

Reply: Accepted and references that are more appropriate were cited here in the revised version:

*"The stems and leaves of submerged macrophytes can adsorb heavy metals from water, and fix*

*pollutants through their roots to improve the quality of lake water bodies (Gumbricht et al., 1993). Also, the outspread of submerged macrophytes would increase the carbon dioxide ($CO_2$) and methane ($CH_4$) release through the methanogenic decomposition of plant exudations and debris (Watanabe et al., 1999; Emilson et al., 2018; Waldo et al., 2019)."*

Line 120 figure 1b should be added after the description of the aquatic macrophytes.
Reply: Accepted and detailed description was added as follows:
*"The submerged macrophytes including Potamogeton pectinatus, Myriophyllum spicatum, and Potamogeton crispus, and emergent plants including Phragmites australis, Typha latifolia. The dominant species are Phragmites australis for emergent vegetation and Potamogeton pectinatus for submerged macrophytes (Duan et al., 2007; Qing et al., 2020, Figure.1b)."*

Line 120 all the species of submerged/emergent macrophytes and their relative proportions should be provided.
Reply: Accepted and added. Also, please see our reply above.

Line 130, why these two types of macrophytes are selected as representations?
Reply: These two types of macrophytes are the dominant species and thus were selected as representations. There are of course uncertainties regarding to the contributions to *n*-alkanes from other types of macrophytes. However, the uncertainties should be neglectable considering that 1) most macrophytes share similar n-alkane distribution pattern and carbon isotopic values, and 2) two types of macrophytes are the dominant species in the specific case of Lake Wuliangsu.

Line 155, please give references for calculating CPI and $P_{aq}$, even the detailed method was given in supplementary material.
Reply: We have included the references as follows:
*"n-Alkane-based $P_{aq}$ (proportion of aquatic plants, Ficken et al., 2000) and CPI (carbon preference index, Eglinton and Hamilton, 1967) proxies were calculated using the following equations provided in the Supplementary Material."*

Line 175, the data 33.73% and 37.08% are from? or why you can get these data?
Reply: Sorry for the mistakes. These values are the results from the samples of submerged macrophytes and emergent plants measured by us. We actually have described the samples and method in preprint manuscript in section 2.2 (lines 130-131) and 2.3 (lines 142-145). We revised this sentence as follows, to avoid this potential misunderstanding:
*"Relatively higher TOC values in the lake sediments than those in the lakeshore and riverine sediments suggest an additional supply of autochthonous materials of higher TOC values (33.73% and 37.08% for submerged macrophytes and emergent plants, see Supplementary Table S1)."*

Line 190, this conclusion should be further evaluated as the *n*-alkanes might be selectively enriched relative to other OM fraction, resulting in a higher abundance in sediments than in biomass
Reply: Considering the uncertainty of *n*-alkanes (analytical uncertainty for concentrations of *n*-alkanes would be less than 5%, see lines 157-158), *n*-alkanes in sample S3 (137.05 µg/g) are similar with or very close to submerged macrophytes (132.75 µg/g). Therefore, we don't think there would

be selectively enriched *n*-alkanes relative to other OM fraction. The similar *n*-alkane concentrations between sample S3 and submerged macrophytes indeed indicate a predominant contribution by submerged macrophytes to this sample.

Why the abundances of short-chain *n*-alkanes were not considered in section 3.2?
Reply: The short-chain *n*-alkanes are not source-oriented (a mixture from algae, submerged macrophytes and terrestrial OC input, Aichner et al., 2010; Liu and Liu, 2016). Therefore, short-chain *n*-alkanes are not the main object of this manuscript. Moreover, the concentrations of the short-chain *n*-alkanes are much lower than mid- and long-chain ones (average values of 3.09 μg/g *vs*. 20.15 μg/g for lake and lakeshore sediments). Even results of these compounds are included, conclusion will not be changed. Therefore, we decided not to include them in section 3.2.

Line 205, our previous works reported the *n*-alkane distribution in surface sediments from 30 lakes in middle and lower Yangtze River basin (Environmental Science and Pollution Research 26, 22472-22484, 2019; Organic Geochemistry 122, 29-40, 2018), you can make a comparison with your work
Reply: We have carefully read these papers recommended, and found out that these papers further support our results. Thank you for that! We have cited literatures here as follows:
*"Such n-alkane distribution patterns are similar to aquatic plants collected from lakes in Qinghai-Tibet Plateau (Aichner et al., 2010; Liu et al., 2015; Liu and Liu, 2016) and middle and lower Yangtze River basin (Zhang et al., 2018, 2019; Yu et al., 2021)."*

Line 235, the first sentence in section 3.4 indicated the isotopic compositions of *n*-alkanes in aquatic macrophytes or sediments?
Reply: They are for all samples including aquatic macrophytes and sediments. We revised this sentence to clarify this issue:
*"The $\delta^{13}C$ values of mid- and long-chain odd n-alkanes of all samples (n-$C_{23}$, n-$C_{25}$, n-$C_{27}$, n-$C_{29}$, and n-$C_{31}$) ranged from −36.6‰ to −24.1‰."*

Line 250, the figure caption indicating sediment samples, but it seems the figure contained macrophyte samples.
Reply: Accepted and revised the caption to:
*"Bar plots and corresponding line graphs showing the distribution of the mid- and long-chain n-alkanes (relative abundance, left axis) and their stable carbon isotopic composition (right axis), for six representative samples from the Lake Wuliangsu, including (a) submerged macrophytes (b) emergent plants (c) riverine soil (d) S3 (e) S6 (f) S13"*

Line 295, the calculation methods stated here should be included in the main text.
Reply: For the purpose of readability of this manuscript, we decided to put all the equations in Supplementary Materials. Actually, all the calculation methods have been described in the main text. Therefore, we feel that our way of dealing the calculation methods stated here would be appropriate. Of course, this is not a scientific issue, so we can compromise a little bit if the reviewers and editor insist.

In 4.2, how about using the gross OM isotope to calculate the contribution from different sources,

as the isotopes of all potential contributors can be easily obtained.

Reply: Please see our first reply to the comments above.

**Additional information:**

Here are references mentioned in this response, and some of them will be included in our revised manuscript:

Ahad J. M. E., Ganesharm, R. S., Bryant, C. L., Cisneros, L. M., Ascough, P., Fallick, A. E., and Slater, G. F.: Sources of *n*-alkanes in an urbanized estuary: insights from molecular distributions and compound-specific stable and radiocarbon isotopes, Mar. Chem., 126, 239–249. https://doi.org/10.1016/j.marchem.2011.06.002, 2011.

Aichner, B., Herzschuh, U., and Wilkes, H.: Influence of aquatic macrophytes on the stable carbon isotopic signatures of sedimentary organic matter in lakes on the Tibetan Plateau, Org. Geochem., 41, 706–718, https://doi.org/10.1016/j.orggeochem.2010.02.002, 2010.

Duan, X. N., Wang X. K., Chen, L., Mu, Y. J., and Ouyang Z. Y.: Methane Emission from Aquatic Vegetation Zones of Wuliangsu Lake, Inner Mongolia. Environmental Sci., 28(3): 455-459.

Emilson, E. J. S., Carson, M. A., Yakimovich, K. M., Osterholz, H., Dittmar, T., Gunn, J. M., Mykytczuk, N. C. S., Basiliko, N. and Tanentzap, A. J.: Climate-driven shifts in sediment chemistry enhance methane production in northern lakes. Nat. Commun., 9, https://doi.org/10.1038/s41467-018-04236-2, 2018.

Gumbricht, T.: 1993. Nutrient removal capacity in submersed macrophyte pond systems in a temperature climate. Ecol. Eng., 2(1), 49–61, https://doi.org/10.1016/0925-8574(93)90026-C, 1993.

Hyun, S., Shin, K. H., Lee, S. C., Chang, S. W., and Nam, S. I.: Terrestrial *n*-alkanes and their carbon isotope records from the Hanon paleo-maar sediment, Jeju Island, Korea: implications for paleoclimate and paleovegetation over the last 35 kyrs, Quat. Int. 441 (Part A), 89–100, https://doi.org/10.1016/j.quaint.2016.08.047, 2017.

Jaffé, R., Mead, R., Hernandez, M. E., Peralba, M. C., and DiGuida, O. A.: Origin and transport of sedimentary organic matter in two subtropical estuaries: a comparative, biomarker-based study. Org. Geochem., 32, 507-526, https://doi.org/10.1016/S0146-6380(00)00192-3, 2001.

Lamb, A. L., Wilson, G. P., Leng, M. J.: A review of coastal palaeoclimate and relative sea-level reconstructions using $\delta^{13}$C and C/N ratios in organic material. Earth Sci. Rev., 75, 29–57, https://doi.org/10.1016/j.earscirev.2005.10.003, 2006.

Mazeas, L., Budzinski, H., Raymond, N.: Absence of stable carbon isotopic fractionation of saturated and polycyclic aromatic hydrocarbons during aerobic bacterial biodegradation. Org. Geochem., 33, 1259–1272, https://doi.org/10.1016/S0146-6380(02)00136-5, 2002.

Ni, M., Liang, X., Hou, L., Li, W., He, C.: Submerged macrophytes regulate diurnal nitrous oxide emissions from a shallow eutrophic lake: A case study of Lake Wuliangsuhai in the temperate arid region of China. Sci. Total Environ., 811, 152451, https://doi.org/10.1016/j.scitotenv.2021.152451, 2022.

Qin, B., Zhou, J., Elser, J. J., Gardner, W. S., Deng, J., and Brookes, J. D.: Water depth underpins the relative roles and fates of nitrogen and phosphorus in lakes. Environ. Sci. Tech., 54 (6), 3191–3198, https://doi.org/10.1021/acs.est.9b05858, 2020.

Qing, S., A, R., Shun, B., Zhao, W. J., Bao, Y. H., Hao, Y. L.: Distinguishing and mapping of aquatic vegetations and yellow algae bloom with landsat satellite data in a complex shallow Lake,

China during 1986–2018. Ecol. Indicat., 106073. https://doi.org/10.1016/j.ecolind.2020.106073, 2020.

Waldo, N. B., Hunt, B. K., Fadely, E. C., Moran, J. J., Neumann, R. B.: Plant root exudates increase methane emissions through direct and indirect pathways. Biogeochemistry 145, 213–234, https://doi.org/10.1007/s10533-019-00600-6, 2019.

Watanabe, A., Takeda, T., and Kimura, M.: Evaluation of origins of $CH_4$ carbon emitted from rice paddies. J. Geophys. Res.: Atmos., 104, 23623–23629, https://doi.org/10.1029/1999jd900467, 1999.

Zhang, Y., Su, Y. L., Liu, Z. W., Kong, L. Y., Yu, J. L., and Jin, M.: Aliphatic hydrocarbon biomarkers as indicators of organic matter source and composition in surface sediments from shallow lakes along the lower Yangtze River, Eastern China. Org. Geochem., 122, 29–40, https://doi.org/10.1016/j.orggeochem.2018.04.009, 2018.

Zhang, Y. D., Su, Y. L., Yu, J. L., Liu, Z. W., Du, Y. X., and Jin, M.: Anthropogenically driven differences in *n*-alkane distributions of surface sediments from 19 lakes along the middle Yangtze River, Eastern China. Environ. Sci. Pollut. Res., 26 (22), 22472–22484, https://doi.org/10.1007/s11356-019-05536-w, 2019.

---

## Author Comment (AC2)

Reply: We thank reviewer #2 for the valuable comments and suggestions on improving the manuscript. We have carefully followed the reviewer's suggestions revising our manuscript (see detailed responses below). We hope our response and the revised manuscript can be a satisfaction for the reviewer. FYI, we quoted line numbers regarding to the preprint manuscript online, and the revision will be updated once the reviewing process was completed.

**General comments**

Lacustrine sequences are important archives for past climate changes and anthropogenic activities. Over the past decades, the molecular and carbon isotope compositions have been widely applied for lacustrine sediments. A quite important prerequisite for the paleo applications is the reasonable source identification and proxy interpretation. This study aimed to quantitatively assess the contribution of allochthonous and autochthonous to the surface sediment of a shallow lake. They collected surface sediments from lake and drainage channel, as well as previously published plant data, and built three end-member models. Data in this study is interesting and the major findings will be surely welcomed by readers from the field of organic geochemistry and paleoclimate. Nonetheless, this version of the paper needs significant improvement in terms of both its language and scientific content before it can be accepted for publication.

I strongly suggest the authors move the model equation to the main text and provide an extensive discussion. Additionally, the definition of $f_{emer}$, $f_{riv}$ and $f_{sub}$ is missing, making it difficult to evaluate the accuracy of model #3, which appears to be too simplistic.

Reply: The $f_{emer}$, $f_{riv}$ and $f_{sub}$ are defined by Model #2 (see Eq. (9) and (12)). In Model #3, we fixed $f_{emer}/f_{riv}$ to the $F_{emer-M1}/F_{riv-M1}$ that calculated from Model #1 (see Eq. (3)–(6)). In this way, the updated Model#3 is derived by introducing the results from Model #1 into Model #2, and therefore is the combination of Model #1 and Model #2. To avoid the potential misunderstanding, we have revised this part as follows:

" $F_{emer} = (1 − F_{sub}) \times F_{emer-M1} / (F_{emer-M1} + F_{riv-M1})$ (16)

$F_{riv} = 1 − F_{emer} − F_{sub}$ (17)

Where $F_{emer-M1}$ and $F_{riv-M1}$ are relative contributions to sedimentary n-$C_{23+25+27+29+31}$ alkanes in Model #1 from emergent plants, and riverine sources (see Eq. (3)–(6)), respectively. $F_{sub}$, $F_{emer}$, and $F_{riv}$ are relative contributions to sedimentary n-$C_{23+25+27+29+31}$ alkanes from submerged macrophytes, emergent plants, and riverine sources calculated from model #3, respectively. "

Furthermore, the representativeness of riverine sediment is a concern since only one sample was analyzed, yet the results indicate a high proportion of riverine OC. The effect of such a major weakness on the uncertainty should be discussed.

Reply: Lake Wuliangsu mainly receives discharge from the Hetao Irrigation District via six irrigation channels from the west bank, with the main channel (also known as the Wujia River) contributing ~90% of the total materials (Yang et al., 2019; Cao et al., 2022). Therefore, we believe riverine samples collected from main channel could roughly represents riverine OC sources. Indeed, the riverine sample shows typical terrestrial signature in terms of distribution pattern and compound-specific [13]C values of n-alkanes. We understand that there might be uncertainties since only one

sample was analyzed, however, we still believe that the error derived from this might be minor. For instance, the estimated CPI values (which is independent with all three evaluation models) from Models #2 and #3 are strongly correlated with the actual CPI values. Also, in Models #2 and #3, we observed a predominate contribution from the autochthonous OC to Lake Wuliangsu (mostly >85%, Fig. 7 and Supplementary Table S4-S5), suggesting that the contribution of riverine OC is minor. Nevertheless, we have mentioned the issue of representativeness of riverine sediment in the section of the materials and methods (line 132) as follows:

*"One riverine surface sediment (<3 cm) was collected at the end section of the main irrigation channel. Considering the fact that the main channel (also known as the Wujia River) contributing ~90% of the total materials to Lake Wuliangsu (Yang et al., 2019; Cao et al., 2022), we believe riverine samples collected from main channel could roughly represents riverine OC sources."*

This manuscript has too many objectives that are overly ambitious yet not well-supported. Additionally, some objectives are not well-aligned with the title of the manuscript.
Reply: We agree with the reviewer and carefully revise this part (lines 97-105) as follows:
*"In this study, we analyzed the distribution patterns and $\delta^{13}C$ values of mid- and long-chain n-alkanes from modern aquatic plants and surface sediments collected from the large and shallow Lake Wuliangsu. We aim to identify the distribution patterns and $\delta^{13}C$ characteristics of n-alkanes from major contributors to the sedimentary OC of Lake Wuliangsu, and develop quantitative/semi-quantitative methods to calculate the relative contributions to sediments from different OC sources based on the end-member mixing models. With the new end-member mixing models, we probed the spatial distribution pattern and the controlling factors of allochthonous and autochthonous OC of Lake Wuliangsu under anthropogenic interferences, which would provide an important reference for sustainable management practices of shallow lakes regionally and globally."*
We hope the revised version can be a satisfaction for the reviewer.

Furthermore, the equation used to estimate the CPI and carbon isotope for each homolog is overly complex, and the idea that the carbon isotope composition of a single alkane is dependent on the percentage in the total *n*-alkanes is strange.
Reply: As the estimated CPI values are calculated for the purpose of testing the validity and uncertainty of these models, the estimated CPI and the actual CPI values should be calculated in the same way. That is the reason why the equations are overly complex, but necessary. For the calculation of $\delta^{13}C_{i\text{-est}}$, it is true that the carbon isotope composition of a single alkane is dependent on the product of the percentage in the total *n*-alkanes from three end member (i.e., $\%C_{x\text{-sub}}$, $\%C_{x\text{-emer}}$, and $\%C_{x\text{-riv}}$) and the relative contributions to sedimentary $n\text{-}C_{23+25+27+29+31}$ alkanes from submerged macrophytes, emergent plants, and riverine sources (i.e. $F_{sub}$, $F_{emer}$, and $F_{riv}$). Therefore, we would like to keep these equations in the manuscript, as they are what they are.

I suggest that the current subsection 4.4 be removed, as it is beyond the scope of source identification and quantification. Moreover, the correlation between TOC and lipid ratio cannot be interpreted as a causal relationship between OC content and bacterial activities. It is more likely that the OC content is the result of heterotrophic decomposition. Furthermore, it is difficult to confidently assert that the lake is anaerobic without providing evidence. These interesting issues

should be explored further, but they should not all be included in one paper without adequate support.

Reply: Actually, section 4.4 is the application for our model results in further evaluation on the previously published tetraether data. The direct comparison between the OC from various sources and the activities of heterotrophic anaerobic microbial can potentially illustrate the relation among the OC input, OC degradation and OC deposition. We understand that the reviewer shows concerns on the GDGT data, but we have provided detailed evidence and reasons why the GDGT-based results represent the activities of heterotrophic anaerobic microbial. Basically, $R_{0/5}$, a diagnostic proxy defined as GDGT-0/crenarchaeol values (Blaga et al., 2009; Ge et al., 2015), can track the activities of methanogenic *Euryarchaeota*, whereas $BR_{in\text{-}situ/soil}$ is a ratio for *in-situ* heterotrophic anaerobic bacteria (*Acidobacteria*) to overall microorganisms in Lake Wuliangsu (He et al., 2023). In the case of Lake Wuliangsu, we observed a strong positive correlation between the OC (both bulk and source-specific ones) and both the $R_{0/5}$ and $BR_{in\text{-}situ/soil}$ ratios. This simply rules out the possibility that the OC content is the result of heterotrophic decomposition, which would generally lead to a negative correlation instead. Therefore, we believe our results indicate high OC content promotes more heterotrophic decomposition, further resulting in stronger microbial degradation. Better correlations with both $R_{0/5}$ and $BR_{in\text{-}situ/soil}$ values in bulk OC and submerged macrophyte-sourced OC than in emergent plant-sourced and riverine-sourced OC also indicate the important role of OC source and amount to heterotrophic microbial activities in Lake Wuliangsu. For the issue of whether Lake is anaerobic, anaerobic condition in Lake Wuliangsu have been observed from previous investigation (Li et al., 2022; Sun et al., 2022; Zhang et al., 2022), especially in open water zone. Furthermore, many studies have suggested that dense stands of submerged macrophytes could provide more suitable substrates for methanogenic archaea to produce $CH_4$ (Xiao et al., 2017; Hilt et al., 2022). All these pieces of evidence support our notion that open water zones are prone to create anaerobic conditions.

For all these reasons, we insist on keep this section, but further polishing this section to clarify the objectives and the main results of this section. For instance, we revised the topic of section 4.4 as follows:

*"Effect of OC source and amount to heterotrophic microbial activities in Lake Wuliangsu."*

We also added extensive discussion as follows (line 494):

*"Positive correlation was observed between TOC values and $R_{0/5}$ values ($r^2$=0.46, p<0.01, n=19) and $BR_{in\text{-}situ/soil}$ ($r^2$=0.66, p<0.01, n=19). When comparing our quantitative evaluation of the submerged macrophytes, emergent plants, and riverine sourced OC values with the $R_{0/5}$ and $BR_{in\text{-}situ/soil}$ ratio, we observe a strong correlation between the submerged macrophyte-sourced OC and both the $R_{0/5}$ and $BR_{in\text{-}situ/soil}$ ratios ($r^2$=0.68 and $r^2$=0.81, p<0.01, n=33, Figure. 8a–b, Supplementary Table S4), and no correlation between emergent plant-sourced OC vs. both the $R_{0/5}$ and $BR_{in\text{-}situ/soil}$ ratios ($r^2$=0.04 and $r^2$=0.10, n=33, Figure. 8c–d), and riverine sourced OC vs. both the $R_{0/5}$ and $BR_{in\text{-}situ/soil}$ ratios ($r^2$=0.02 and $r^2$=0.07, n=33, Figure. 8c–d). Better correlations with both $R_{0/5}$ and $BR_{in\text{-}situ/soil}$ values in bulk OC and submerged macrophyte-sourced OC than in emergent plant-sourced and riverine-sourced OC also indicate the important role in both OC input and OC degradation."*

The introduction section lacks a clear logic, making it difficult for readers to understand the advances and knowledge gaps in this field from a broad perspective. The current version is too narrow and hard to comprehend.

Reply: In the introduction section, we firstly talked about why knowledge of OC burial behavior of Lake Wuliangsu under anthropogenic interferences would provide an important reference for sustainable management practices of shallow lakes regionally and globally. Then, we mentioned about the necessarily of evaluating OC contributions from allochthonous (terrestrial input) and autochthonous (primary bioproductivity) sources in shallow lake systems, considering difference types of OC have their own characteristics on perspectives of source and sink behavior. After that, we reviewed the previous studies within OC identification in Lake Wuliangsu and illustrate the advantages of the proxies derived from distribution pattern and compound-specific $^{13}C$ of $n$-alkanes. In the end, we pointed out our objectives, which are to identify the distribution patterns and $\delta^{13}C$ characteristics of $n$-alkanes from major contributors to the sedimentary OC of Lake Wuliangsu, and develop quantitative/semi-quantitative methods to calculate the relative contributions to sediments from different OC sources based on the end-member mixing model.

As the largest lake ecosystem in the reaches of the Yellow River, Lake Wuliangsu is a typical semi-arid shallow lake that has undergone strong eutrophication and paludification due to rapid industrialization and urbanization over the last decades. We believe that the results from Lake Wuliangsu would provide an important reference for sustainable management practices of shallow lakes regionally and globally.

In the revised manuscript, we went through the whole Introduction section and polished the language and logic at our best. We hope the revision can be a satisfaction for you.

**Major changes**

Mistakes are frequently found in this document. Here are some examples.

The sample number labeled in Fig. 1 is inconsistent with the text and the other figures.

Reply: Thank you for pointing this out, and we are very sorry for the trouble. We have revised the samples number in Fig. 1.

In addition, the rationale to distinguish the offshore lake sediments and other lake sediments is absent.

Reply: We designed two types of samples for several reasons: Firstly, the locations of these two types of samples was totally different, lake sediments were collected via boating in central lake, while lakeshore sediments were collected via walking into the lakeshore by foot. Secondly, the $CPI_{est}$ values for lake sediments show positive correlation with actual CPI values ($r^2$=0.59, n=14, $p$<0.01), those for lakeshore sediments show very weak but negative correlation with actual CPI values ($r^2$=0.16, n=6). Furthermore, distinguishing lakeshore sediments and lake sediments would help better represent the difference in OC values derived from different OC sources in Fig. 7. Therefore, we decide to distinguish the offshore lake sediments and other lake sediments.

We have added these sentences in section 2.2 (line 126) as follows:
*"Lake sediments were collected via boating in central lake, while lakeshore sediments were collected from bench segments around the lake."*

L90-91: please check the data in Collister et al (1994). In addition, here you need to tell the results of *n*-alkanes rather than the bulk OM. Other reference citation should be also checked. In the supplement, you cited Eglinton and Hamilton (1967) for the calculation of CPI. This paper did not discuss CPI.

Reply: Accepted and revised this sentence (lines 90-91) as follows:

"The $\delta^{13}C$ values of n-alkanes are dependent on the natures of photosynthesis pathways of the sourced terrestrial plants ($C_3$ vs. $C_4$, −39.0‰ to −32.0‰ vs. −25.0‰ to −18.0‰, Collister et al., 1994)."

Also, we have carefully discussed CPI values both in main text (section 3.3 and 4.2) and supplementary materials, and cited Eglinton and Hamilton (1967) in both of them.

L125: specify the 3-cm as the depth. Similar problem occurs in L129.
Reply: Accepted and added.

L142-143: carbonate
Reply: Accepted and added.

Figure 3: Citation for the plant data is required. In addition, please tell the plant numbers and show the error bar at least for a and b.

Reply: Plant data in this manuscript was collected and measured by us (we have mentioned this in the Materials and Methods section). Considering the fact that *Potamogeton pectinatus* and *Phragmites communis* are dominated species from submerged macrophytes and emergent plants, respectively. We select two samples in submerged macrophytes and one sample in emergent plants to represent autochthonous OC signals (see line 190). To avoid the misunderstanding, we have added the plants number and error bar in Figure. 3(a) and (b).

L541: delete 'and' after Y.H.
Reply: Accepted and added.

**Additional information:**

Here are references mentioned in this response, and some of them will be included in our revised manuscript:

Blaga, C. I., Reichart, G. J., Heiri, O., and Sinninghe Damsté, J. S.: Tetraether membrane lipid distributions in water-column particulate matter and sediments: a study of 47 European lakes along a north–south transect, J. Paleolimnol., 41, 523–540, https://doi.org/10.1007/s10933-008-9242-2, 2009.

Cao, C. M., Li, N., Yue, W. F., Wu, L. J., Cao, X. Y., and Zhai, Y. Z.: Analysis of the Interaction between Lake and Groundwater Based on Water-Salt Balance Method and Stable Isotopic Characteristics, Int. J. Environ. Res. Public Health, 19, 12202. https://doi.org/10.3390/ijerph191912202, 2022.

Collister, J. W., Rieley, G., Stern, B., Eglinton, G., and Fry, B.: Compound-specific $\delta^{13}C$ analyses of leaf lipids from plants with differing carbon dioxide metabolisms, Org. Geochem., 21, 619–627, https://doi.org/10.1016/0146-6380(94)90008-6, 1994.

Eglinton, G., and Hamilton, R. J.: Leaf epicuticular waxes, Science, 156, 1322–1335,

https://doi.org/10.1126/science.156.3780.132, 1967.

Ge, H. M., Zhang, C. L., Versteegh, G. J. M., Chen, L. L., Fan, D. D., Dong, L., and Liu, J. J.: Evolution of the East China Sea sedimentary environment in the past 14 kyr: Insights from tetraethers-based proxies, Sci. China Earth Sci., 59, 927–938, https://doi.org/10.1007/s11430-015-5229-9, 2015.

He, Y., Zhao, Q., and Sun, D.: *In-situ* activities of anaerobic bacteria and methanogenic archaea in shallow lakes over the Anthropocene: A case study of Lake Wuliangsu, Chem. Geol., 619, 121312, https://doi.org/10.1016/j.chemgeo.2023.121312, 2023.

Hilt, S., Grossart, H. P., McGinnis, D. F., and Keppler, F.: Potential role of submerged macrophytes for oxic methane production in aquatic ecosystems, Limnol. Oceanogr., 67, S76–S88, https://doi.org/10.1002/lno.12095, 2022.

Li, G., Zhang, S., Shi, X., Zhan, L., Zhao, S., Sun, B., Liu, Y., Tian, Z., Li, Z., Arvola, L., Uusheimo, S., Tulonen, T., and Huotari, J.: Spatiotemporal variability and diffusive emissions of greenhouse gas in a shallow eutrophic lake in Inner Mongolia, China, Ecol. Indic., 145, 109578, https://doi.org/10.1016/j.ecolind.2022.109578, 2022.

Sun H., Yu, R., Liu, X., Cao, Z., Li, X., Zhang, Z., Wang, J., Zhuang, S., Ge, Z., Zhang, L., Sun, L., Lorke, A., Yang, J., Lu, C., and Lu, X.: Drivers of spatial and seasonal variations of $CO_2$ and $CH_4$ fluxes at the sediment water interface in a shallow eutrophic lake, Water Res., 222, 118916, https://doi.org/10.1016/j.watres.2022.118916, 2022.

Xiao, Q. T., Zhang, M., Hu, Z. H., Gao, Y. Q., Hu, C., Liu, C., Liu, S. D., Zhang, Z., Zhao, J. Y., Xiao, W., and Lee, X.: Spatial variations of methane emission in a large shallow eutrophic lake in subtropical climate, J. Geophys. Res.-Biogeosci., 122(7): 1597-1614, https://doi.org/10.1002/2017JG003805, 2017.

Yang, Y., Weng, B., Bi, W., Xu, T., Yan, D., and Ma, J.: Climate Change Impacts on Drought-Flood Abrupt Alternation and Water Quality in the Hetao Area, China, Water, 11, 652, https://doi.org/10.3390/w11040652, 2019.

Zhang, F., Shi, X., Zhao, S., Hao, R., and Zhai, J.: Equilibrium analysis of dissolved oxygen in Lake Wuliangsuhai during ice-covered period, J. Lake Sci., 34, 1570–1583, https://doi.org/10.18307/2022.0513, 2022.